# Multi-decadal Hydrologic Change and Variability in the Amazon River Basin: Understanding Terrestrial Water Storage Variations and Drought Characteristics

Suyog Chaudhari[1], Yadu Pokhrel[1*], Emilio Moran[2], Gonzalo Miguez-Macho[3]

[1]Department of Civil and Environmental Engineering, Michigan State University, East Lansing, MI 48824
[2]Department of Geography, Environment and Spatial Sciences, Michigan State University, East Lansing, MI 48824
[3]Non-Linear Physics Group, Faculty of Physics 15782, Universidade de Santiago de Compostela, Galicia, Spain.

*Correspondence to*: Yadu Pokhrel (ypokhrel@egr.msu.edu)

**Abstract.** We investigate the interannual and interdecadal hydrological changes in the Amazon River basin and its sub-basins during 1980-2015 period using GRACE satellite data and a physically-based, 2-km grid continental-scale hydrological model (Leaf-Hydro-Flood) that includes a prognostic groundwater scheme and accounts for the effects of land use land cover (LULC) change. The analyses focus on the dominant mechanisms that modulate terrestrial water storage (TWS) variations and droughts. We find that (1) the model simulates the basin-averaged TWS variations remarkably well, however, disagreements are observed in spatial patterns of temporal trends, especially for the post-2008 period, (2) the 2010s is the driest period since 1980, characterized by a major shift in decadal mean compared to 2000s caused by increased drought frequency, (3) long-term trends in TWS suggests that the Amazon overall is getting wetter (1.13 mm/yr), but its southern and southeastern sub-basins are undergoing significant negative TWS changes, caused primarily by intensified LULC changes, (4) increasing divergence between dry season total water deficit and TWS release suggest a strengthening dry season, especially in the southern and southeastern sub-basins, and (5) the sub-surface storage regulates the propagation of meteorological droughts into hydrological droughts by strongly modulating TWS release with respect to its storage preceding the drought condition. Our simulations provide crucial insight on the importance of sub-surface storage in alleviating surface water deficit across Amazon and open pathways for improving prediction and mitigation of extreme droughts under changing climate and increasing hydrologic alterations due to human activities (e.g., LULC change).

## 1. Introduction

The Amazon River basin is one of the most hydrologically and ecologically diverse regions in the world (Fan and Miguez-Macho, 2010; Latrubesse et al., 2017; Lenton et al., 2009; Lesack, 1993; Malhi et al., 2008; Moran et al., 2018; Timpe and Kaplan, 2017; Tófoli et al., 2017). It is home to the world's largest tropical rainforest and hosts ~25% of all terrestrial species on Earth (Malhi et al., 2008). Hydrologically, it contributes to 20-30% of the world's total river discharge into the oceans (Clark et al., 2015; Muller-Karger et al., 1988; Nepstad et al., 2008) and accounts for ~15% of global terrestrial

evapotranspiration (Field et al., 1998; Malhi et al., 2008). Thus, the Amazon is an important component of global terrestrial ecosystems and the hydrologic cycle (Cox et al., 2004; Nobre et al., 1991); it also plays a major role in global atmospheric circulation through precipitation recycling and atmospheric moisture transport (Malhi et al., 2008; Soares-Filho et al., 2010).

The hydro-ecological systems of the Amazon are dependent on plentiful rainfall (Cook et al., 2012; Espinoza et al., 2015, 2016; Espinoza Villar et al., 2009; Nepstad et al., 2008) and the vast amount of water that flows down through extensive river networks and massive floodplains (Bonnet et al., 2008; Coe et al., 2002; Frappart et al., 2011; Miguez-Macho and Fan, 2012a; Yamazaki et al., 2011; Zulkafli et al., 2016). The spatiotemporal patterns of precipitation are, however, changing due to climate change and variability (Brando et al., 2014; Cook et al., 2012; Lima et al., 2014; Malhi et al., 2008, 2009; Nepstad et al., 2008), large-scale alterations in land use (e.g., deforestation) (Chen et al., 2015; Coe et al., 2009; Davidson et al., 2012; Kalamandeen et al., 2018; Lima et al., 2014; Panday et al., 2015; Tollefson, 2016), and more recently the construction of mega-dams (Finer and Jenkins, 2012; Latrubesse et al., 2017; Moran et al., 2018; Soito and Freitas, 2011; Timpe and Kaplan, 2017; Winemiller et al., 2016), among others. Such changes in precipitation patterns typically manifest themselves in terms of altered magnitude, duration, and timing of streamflow (Marengo, 2005). A prominent streamflow alteration pattern that has been widely observed across the Amazon is the extended dry-season length (Espinoza et al., 2016; Marengo et al., 2011) and an increase in the number of dry events (i.e., droughts) over the longer term (Malhi et al., 2009; Marengo and Espinoza, 2016), which has been suggested to be a result of ongoing climatic and human-induced changes (Cook et al., 2012; Cook and Vizy, 2008; Lee et al., 2011; Malhi et al., 2008; Shukla et al., 1990). However, the cross-scale interactions and feedbacks in the human-water relationship make it difficult to explicitly quantify the causes. These changes have resulted in decreases in runoff (Espinoza et al., 2009; Haddeland et al., 2014; Lima et al., 2014), and loss of terrestrial biodiversity (Barletta et al., 2010; Newbold et al., 2016; Tófoli et al., 2017; Toomey et al., 2011; Winemiller et al., 2016). Increased variability in streamflow has also resulted in the disruption of the food pulse and fishery yields, which the Amazon region thrives upon (Castello et al., 2013, 2015; Forsberg et al., 2017). Moreover, persistent dry events create social negative externalities, such as deterioration of respiratory health due to drought induced fires (Smith et al., 2014), exhaustion of family savings (Brondizio and Moran, 2008), isolation of communities that are affected by navigation and drinking water scarcity (Sena et al., 2012), hence affecting the overall livelihood of the local communities. Thus, it is critical to understand the characteristics of the historical droughts to better understand the dominant mechanisms that modulate droughts and their evolution over time.

As often is the case, droughts in the Amazon are driven by El Niño events, however, some droughts are suggested to be caused by climate change and variability (Espinoza et al., 2011; Lewis et al., 2011; Marengo et al., 2008; Marengo and Espinoza, 2016; Phillips et al., 2009; Xu et al., 2011; Zeng et al., 2008) and due to accelerating activities causing rapid changes in land use/water cycle (Lima et al., 2014; Malhi et al., 2008). Numerous studies have quantified the impacts and spatial extent of these periodic droughts on the hydrological and ecological systems in the Amazon (Alho et al., 2015; Brando et al., 2014; Castello et al., 2013, 2015, Chen et al., 2009, 2010; da Costa et al., 2010; Davidson et al., 2012; Fernandes et al., 2011; Lewis et al., 2011; Phillips et al., 2009; Saleska et al., 2007, 2016; Satyamurty et al., 2013; Schöngart and Junk, 2007; Xu et al., 2011; Zeng et al., 2008). For example, Lewis et al., (2011) found that the 2010 drought was spatially more extensive than the 2005

drought; the spatial extent was over 3.0 million km$^2$ in 2010 and 1.9 million km$^2$ in 2005. These catastrophic droughts had major implications on the hydrology of the Amazon River basin; for example, the 2005 hydrological drought led to reduction in streamflow by 32% from the long-term mean, as reported in Zeng et al., (2008), and in 2010 moisture stress induced persistent declines in vegetation greenness affecting an area of ~2.4 million km$^2$ which was 4 times greater than the area impacted in 2005 (Xu et al., 2011). Moreover, these extreme drought events, coupled with forest fragmentation have caused widespread fire-induced tree mortality and forest degradation across Amazonian forests (Aragão et al., 2007; Brando et al., 2014; Davidson et al., 2012; Malhi et al., 2008; Rammig et al., 2010).

Due to the limited availability of observed data (e.g., precipitation, streamflow) for the entire basin, hydrologic characteristics of droughts in the Amazon has been studied primarily by using hydrological models and satellite remote sensing. For example, early studies (Coe et al., 2002; Costa and Foley, 1999; Lesack, 1993; Vorosmarty et al., 1996; Zeng, 1999) examined different components of the Amazon water budget and their trends through relatively simpler models. More recent literature (Dias et al., 2015; Fan et al., 2019; Getirana et al., 2012; Miguez-Macho and Fan, 2012a, 2012b, Paiva et al., 2013b, 2013a, Pokhrel et al., 2012b, 2012a, 2013; Shin et al., 2018; Siqueira et al., 2018; Wang et al., 2019; Yamazaki et al., 2011, 2012) provided further advances in modeling the hydrological dynamics connected with anthropogenic activities in the Amazon and other parts of the world. Methods with varying complexities were used in similar studies, ranging from simple water budget analyses, (Betts et al., 2005; Costa and Foley, 1999; Fernandes et al., 2008; Lesack, 1993; Sahoo et al., 2011; Vorosmarty et al., 1996; Zeng, 1999) to state-of-the-art land surface models (Getirana et al., 2012; Miguez-Macho and Fan, 2012a, 2012b, Paiva et al., 2013a, 2013b; Pokhrel et al., 2013; Siqueira et al., 2018; Wongchuig Correa et al., 2017; Yamazaki et al., 2011, 2012), with some targeting the overall development of parameterization and process representation in the model (Coe et al., 2008, 2009; Dias et al., 2015; Getirana et al., 2010, 2012, Miguez-Macho and Fan, 2012a, 2012b; Paiva et al., 2013b; Pokhrel et al., 2013; Yamazaki et al., 2011), and others on the hydrological changes occurring in the basin due to weather variability (Coe et al., 2002; Lima et al., 2014; Wongchuig Correa et al., 2017).

The major droughts events in the Amazon, particularly those in recent years, have been detected by satellite remote sensing and their impacts on terrestrial hydrology have been examined (Chen et al., 2010; Filizola et al., 2014; Xu et al., 2011). In particular, the hydrologic impact of droughts have been revealed by examining the anomalies in terrestrial water storage (TWS) inferred from the Gravity Recovery and Climate Experiment (GRACE) satellites. A significant decrease in TWS over Central Amazon in the summer of 2005, relative to the average of the five other summer months during 2003-2007 period, was reported by Chen et al., 2009. However, due to the vast latitudinal extent of the Amazon basin, these severe dry conditions were observed only in some regions of the basin. Xavier et al., (2010) and Frappart et al., (2013) used GRACE TWS estimates to identify the signature of these drought events and suggested that the 2005 drought only affected the western and central parts of the basin, whereas very wet conditions peaking in mid-2006 were observed in the eastern, northern and southern regions of the basin. Although the ramifications of these extreme droughts have been widely studied using remote sensing datasets (e.g., GRACE), the understanding of their time-evolution is limited due to data gaps and short study periods, hence hindering their comprehensive categorization. Further, GRACE provides the changes in vertically integrated TWS variations, thus variations

in the individual TWS components cannot be estimated solely by GRACE. This shortcoming is overcome by using hydrological models that separate TWS into its individual components and provide simulations for an extended timescale. However, discrepancy between models and GRACE observations has also become a major topic of discussion, as most of the global models show an opposite trend in TWS compared to GRACE in Amazon and other global river basins (Scanlon et al.,

2018); yet, no clear explanation or quantification exist in the published literature, apart from the attribution of the discrepancy to model shortcomings (see Section 3.3 for details).

As referenced above, the changing hydro-climatology of the Amazon basin, along with specific drought related analysis (e.g. 2005, 2010) has been widely reported in a large body of literature published over recent decades. Several studies have used statistical measures to quantify drought severity (Espinoza et al., 2016; Gloor et al., 2013; Joetzjer et al., 2013; Marengo, 2006;

Marengo et al., 2008, 2011; Wongchuig Correa et al., 2017; Zeng et al., 2008; Zhao et al., 2017a), concerning common variables, such as streamflow and precipitation, thus limiting the quantification of drought impact on water stores viz. flood, groundwater and TWS. Further, even though these studies encompass different aspects of hydrological and climatic changes, most span over only a few years to a decade, except for some precipitation related studies (Marengo, 2004; Marengo et al., 1998). Other studies have used a relatively longer study period (Costa et al., 2003; Espinoza et al., 2016; Zeng, 1999), but the

spatial extent is limited. Thus, a comprehensive understanding of the interdecadal hydrologic change and variability across the entire basin and that of changes in drought characteristics is still lacking. Given the number of droughts that have occurred and their widespread impact in the Amazon, it is imperative to have a better understanding of these past events so as to anticipate future hydrological conditions (Phipps et al., 2013). Many aspects of the droughts are yet to be studied, such as, the interdependence between TWS and meteorological (precipitation-related) and hydrological (streamflow-related) droughts. A

complete categorization of the drought events with respect to their causes and impacts and the resulting basin response is still coming up short.

In this study, we investigate the interannual and interdecadal variability in TWS and drought events in the Amazon River basin over 1980-2015 period. Our study is driven by the following key science questions: (1) how do interannual and interdecadal changes in drought conditions manifest as long-term variations in TWS at varying spatial and temporal scales in the Amazon

River basin? (2) What are the impacts of TWS variations on dry season water deficit and release? Is the Amazonian dry season getting stronger/severe? (3) what are the dominant factors driving the evolution of TWS and drought conditions at varying spatial and temporal scales? And (4) how does the sub-surface water storage regulate the water deficiency caused by the surface drought conditions? These questions are answered by using hydrological simulations from a continental-scale hydrological model and the TWS data from GRACE satellites; the goal is to provide a comprehensive picture of characteristics and evolution

of droughts in the Amazon with respect to their types and spatial impact. Specifically, this study aims to: i) examine the impacts of drought conditions on TWS and other hydrological variables; ii) understand the hydrological variability and drought evolution in the Amazon at an annual and decadal scale over the past four decades; iii) quantify the role of sub-surface water storage in alleviating the surface drought conditions; and iv) summarize each drought year by providing a comprehensive characterization for the major drought events in the Amazon and its sub-basins.

## 2. Model and Data

### 2.1 The Leaf-Hydro-Flood (LHF) Model

The model used in this study is LHF (Fan et al., 2013; Miguez-Macho and Fan, 2012b, 2012a, Pokhrel et al., 2013, 2014), a continental-scale land hydrology model that resolves various land surface hydrologic and groundwater processes on a full physical basis. It is derived from the model Land-Ecosystem-Atmosphere Feedback (LEAF) (Walko et al., 2000), the land surface component of the Regional Atmosphere Modeling System (RAMS) (Pielke et al., 1992). The original LEAF was extensively improved and enhanced to develop LEAF-Hydro for North America (Fan et al., 2007; Miguez-Macho et al., 2007) by adding a prognostic groundwater storage and allowing (1) the water table to rise and fall or the vadose zone to shrink or grow, (2) the water table, recharged by soil drainage, to relax through streamflow into rivers, and lateral groundwater flow, leading to convergence to low valleys, (3) two-way exchange between groundwater and rivers, representing both losing and gaining streams, (4) river routing to the ocean as kinematic waves, and (5) setting sea level as the groundwater head boundary condition. Miguez-Macho and Fan, (2012a) further enhanced the LEAF-Hydro framework by incorporating the river-floodplain routing scheme which solves the full momentum equation of open channel flow, giving more realistic streamflow estimates by considering the prominent backwater effect observed in the Amazon (Bates et al., 2010; Yamazaki et al., 2011). LHF model has been extensively validated in the North and South American continents at 5km and 2km grids, respectively (Fan et al., 2013; Miguez-Macho et al., 2008; Miguez-Macho and Fan, 2012a, 2012b; Pokhrel et al., 2013; Shin et al., 2018) and used to examine the impacts of climate change on groundwater system in the Amazon (Pokhrel et al., 2014). A complete description of LHF can be found in Miguez-Macho and Fan (2012a).

### 2.2 Atmospheric Forcing

Atmospheric forcing data are taken from WATCH Forcing Data methodology applied to ERA-Interim reanalysis data (WFDEI) (Weedon et al., 2014), available for the 1979-2016 period at 0.5º spatial resolution and 3-hr timesteps. WFDEI dataset is widely used in for both global and regional scales studies (Beck et al., 2016; Felfelani et al., 2017; Hanasaki et al., 2018; Schmied et al., 2014), and has been suggested to well represent the observations in the Amazon region (Monteiro et al., 2016). The original WFDEI data at 0.5º resolution are spatially interpolated using a bilinear interpolation method to model grid resolution (~2km), following our previous studies (Miguez-Macho and Fan, 2012a, 2012b, Pokhrel et al., 2013, 2014; Shin et al., 2018). The more recent European Centre for Medium-Range Weather Forecasts Reanalysis 5[th] (ERA5) dataset, which provides atmospheric forcing data from 1979 to present day at a spatial resolution of 0.25º, show promise by outperforming its predecessors (Towner et al., 2019). However, as no studies existed in the past literature which comprehensively validated the ERA5 dataset over the Amazon region until recently, WFDEI forcing remains a better alternative as a model input.

## 2.3 Land Use Land Cover and Leaf Area Index

The land cover data used in this study are obtained from the European Space Agency Climate Change Initiative's Land Cover project (ESA-CCI; http://maps.elie.ucl.ac.be/CCI/). The data comprise of an annual timeseries of high-resolution land cover maps for 1992-2015 period at a 300m spatial resolution, generated by combining the baseline map from the Medium-spectral Resolution Imaging Spectrometer (MERIS) instrument and the land use land cover (LULC) changes detected from AVHRR (1992 - 1999), SPOT-Vegetation (1999 - 2012) and PROBA-V (2013 - 2015) instruments. The classification follows the LULC classes defined by the UN Land Cover Classification System (LCCS). Spatiotemporal coverage and resolution of these LULC maps are consistent with the specific LHF model requirements; hence we use annual land cover input, spatially aggregated to 2km LHF model grids, following the general practice in hydrologic impact studies (Arantes et al., 2016; Panday et al., 2015). Because the ESA-CCI data did not cover the simulation period prior to year 1992, we derive the time-series products for 1980-1991 period by using the trend in leaf area index (LAI) and the ESA-CCI landcover map for year 1992 as a baseline. A pixel-by-pixel analysis is conducted and the pixels with mean annual LAI higher than 5 are transitioned into forest canopy, whereas for other pixels LULC type is retained from the previous year's LULC map. The threshold of LAI equal to 5 for facilitating the land cover transition into forest is determined based on the LAI classifications provided in past literature (Asner et al., 2003; Myneni et al., 2007; Xu et al., 2018). Reverse prediction of LULC changes was constrained to forest canopy only, as it is difficult to predict the LULC type based on LAI values less than 5. Also, forest cover is known to be the most prominent land cover in the Amazon, hence it is reasonable to assume that most of the LULC changes occurring in the basin are transitioned from forest cover.

Monthly LAI data are derived by temporally aggregating the 8-day composites from Global Land Surface Satellite (GLASS) LAI product (Liang and Xiao, 2012; Xiao et al., 2014) to monthly values for the entire model domain. GLASS LAI values for the period of 1982-1999 are derived from AVHRR reflectance, whereas MODIS reflectance values are used for period 2000-2012. Because of the data constraint, LAI data for years before 1982 and after 2012 are assumed to be the same as that of years 1982 and 2012, respectively.

## 2.4 Validation Data

### 2.4.1 Observed Streamflow

We use monthly averaged streamflow data obtained from the Agência Nacional de Águas (ANA) in Brazil (http://hidroweb.ana.gov.br). Fifty-five stream gauge stations are selected considering a wide coverage over the Amazonian sub-basins, and a good balance between low and high flow values. The major selection criterion is the data length; i.e., we only include gauges with at least 30 years coverage. In a few cases, such as for Japura sub-basin, the threshold was overlooked because this criterion resulted in a small number of gauging stations. All the selected stations have observational data for varying time frames with minimal data gaps; the months with missing data are skipped in the statistical analysis.

### 2.4.2 GRACE Data

The TWS products from the GRACE satellite mission are used to validate the TWS simulated by LHF for 2002-2015 period. Equivalent water height from three processing centers, namely: (i) Jet Propulsion Laboratory (JPL) , (ii) the Center for Space Research (CSR), and (iii) the German Research Center for Geoscience (GFZ) (http://grace.jpl.nasa.gov/data/ get-data/) (Landerer and Swenson, 2012) are used along with two mascon products from CSR and JPL; mascon products have been suggested to better capture TWS signals in many regions (Scanlon et al., 2016). Basin-averaged data of variation in TWS anomalies are calculated from GRACE by taking an area-weighted arithmetic mean with varying cell area (Felfelani et al., 2017).

### 2.5 TWS Drought Severity Index

To examine the occurrence and severity of hydrological droughts over the past decades, we employ the drought severity index derived from time-varying TWS change from GRACE, known as the GRACE Drought Severity Index (GRACE-DSI) (Zhao et al., 2017b). We apply GRACE-DSI framework to the 36-year simulated TWS (referred hereafter to as TWS-DSI) to examine the interannual and interdecadal drought evolution over the entire basin. This index is solely based on the TWS anomalies and has been shown to capture the past droughts with favorable agreement with other drought indices derived from precipitation (e.g., PDSI and SPEI) (Zhao et al., 2017b, 2017a). TWS-DSI is calculated for each grid cell in the model domain as follows,

$$TWS\_DSI_{i,j} = \frac{TWS_{i,j} - \overline{TWS}_j}{\sigma_j}, \tag{1}$$

where, $TWS_{i,j}$ is the TWS anomaly from LHF for year $i$ and month $j$; and $\overline{TWS}_j$ and $\sigma_j$ are the temporal mean and standard deviation of TWS anomalies for month $j$, respectively.

### 2.6 Occurrence and Duration of Drought

The characteristics of hydrological droughts are identified from the simulated streamflow using the widely used threshold level approach. Different thresholds have been proposed in previous studies: mean flow, minimum and maximum flows (Marengo and Espinoza, 2016; Wongchuig Correa et al., 2017), 80[th] percentile ($Q_{80}$) flow (Van Loon et al., 2012; Van Loon and Laaha, 2015; Wanders and Van Lanen, 2015), and 90[th] percentile ($Q_{90}$) flow (Wanders et al., 2015; Wanders and Wada, 2015). In this study, we use $Q_{90}$ which is derived from the flow duration curve where $Q_{90}$ is the streamflow that is equaled or exceeded for 90% of the time. $Q_{90}$ is used to isolate severe drought events over the simulation period. Monthly threshold values are derived using the 36-year simulated streamflow and are smoothed by a 30-day moving average. Drought condition is identified by determining whether the variable is below the threshold, expressed mathematically as,

$$Ds(t,x) = \begin{cases} 1 & for\ Q(t,x) < Q_{90}(t,x) \\ 0 & for\ Q(t,x) \geq Q_{90}(t,x) \end{cases}, \tag{2}$$

where $Ds(t,x)$ indicates whether the grid $(x)$ is in a drought state at time $(t)$, $Q(t,x)$ is the streamflow and $Q_{90}(t,x)$ is the threshold for grid $(x)$ at time $(t)$. Consecutive drought states are added to get the drought duration. Events with duration less

than 3 days are not considered as droughts. The number of drought days per year is calculated by aggregating the duration of all the drought events in a year.

## 2.7 Dry Season Total Water Deficit

We define the dry season total water deficit (TWD) as the cumulative difference between monthly potential evapotranspiration (PET) and precipitation (P) for the period during which P<PET. The corresponding drop in the simulated TWS, during the same period as of TWD, is defined as the TWS release (TWS-R). TWD and TWS-R can be conceptualized as the annual water demand and supply as described in Guan et al., (2015). PET estimated at the daily interval using the Penman Monteith approach (Monteith, 1965) as in Pokhrel et al., (2014) is aggregated to the monthly scale to calculate TWD; for consistency, we use the WFDEI forcing data that is used for LHF simulations (section 2.2). TWS anomalies required for the estimation of storage release are obtained from the LHF model.

## 2.8 Simulation Setup

LHF is setup for the entire Amazon basin (~7.1 million km$^2$) including the Tocantins River Basin. Simulations are conducted for the 1979-2015 period at a spatial resolution of 1 arc minute (~2 km). Model time step is 4 minutes as in previous studies (Miguez-Macho and Fan, 2012b, 2012a, Pokhrel et al., 2013, 2014), however, model output is saved at daily timesteps. To stabilize water table depth, the model is spun up for ~150 years starting with the equilibrium water table (Fan et al., 2013) for 1979 and results for 1980-2015 period (36 years) are analyzed. As this study aims to analyze the hydrological changes in Amazon on a decadal scale, simulations for 1979 are considered as additional spin-up and hence not used. Dynamic monthly LAI and annual LULC maps are used to account for LULC changes (see Sections 2.3). Moreover, as the model simulates land surface, hydrologic and groundwater processes on a complete physical basis, no calibration was performed on the model output. Original novelty of the LHF model framework, combined with the incorporated dynamic human role through land cover change creates a "state-of-the-art" framework for assessing long-term hydrological changes. Complete LHF framework along with the input data employed in this study is presented in Figure S1.

## 3. Results and Discussion

### 3.1 Evaluation of Simulated Streamflow

Figure 1 presents the Taylor diagram (Taylor, 2001) illustrating the statistics of the simulated streamflow against observations at 55 gauging locations (see Section 2.4.1 and Figure S2) across the entire Amazon basin. The Taylor diagram provides a synthetic view of error in the simulations in terms of the ratio of standard deviation (SD) of the simulated streamflow to the observed as a radial distance and their correlation as an angle in the polar axis. Most of the stations show a high correlation (> 0.8) and a SD ratio close to unity, indicating a good model performance overall for varying geographical locations and stream sizes over the Amazon. Low correlation (~0.6) is seen for some gauging stations situated on streams with smaller annual mean

flow and steep slope profile; for example, the smaller streams across the Andes in Japura and Negro sub-basins, along with the streams in northeastern parts of Amazon. In these streams with high topographic gradients, precipitated water quickly flows away causing slightly erratic patterns of seasonal streamflow, which is apparent in both simulated and observed timeseries (Figures S3 and S4). However, due to the difficulty in resolving hillslopes processes for low-order streams using 2km grids, the model is unable to fully capture the flow seasonality in the streams with high topographic gradient.

The spatial distribution of the simulated streamflow across the entire model domain and the timeseries comparison of simulated vs. observed streamflow at 12 selected stations are presented in the SI (Figures S2 and S3). The simulated seasonal cycle compares well with the observed one for the entire basin (i.e., Obidos station) as well as for most sub-basins; however, discrepancies in the seasonal peaks can be seen in some basins (e.g., Xingu, Tocantins, and Tapajos). Man-made reservoirs generally attenuate streamflow peaks and seasonal variability, reducing the SD, which is reflected in the observed data but not yet accounted for in the model; this could have exaggerated the SD ratio in some cases. For example, the streamflow in the Tocantins River shows higher SD compared to observed streamflow, likely due to the operation of the Tucurui I and II dams. Conversely, the SD ratio is lower than unity at some stations, including those in the Madeira River (Figure 1), due to the dry bias found in the input precipitation (see Figure S5 and Section 3.2). For sub-basins with higher groundwater contribution to streamflow, such as Xingu, Tapajos, Tocantins and Madeira, the dry-season flow is overestimated (Figure S3), which results from possibly exaggerated groundwater buffer in the model for these regions (Miguez-Macho and Fan, 2012a). Given that LHF is a continental-scale model, simulates streamflow on a full physical basis, and is not calibrated with observed streamflow, we consider these results to be satisfactory to study the hydrologic changes and variability.

**3.2 Evaluation of Simulated TWS Anomalies with GRACE**

Figure 2 presents the comparison of simulated TWS anomalies and GRACE data for the entire Amazon basin and its eight sub-basins; for model results, the individual TWS components are also provided. The model performs very well in simulating the basin averaged TWS anomalies for the entire Amazon basin and most sub-basins. However, some difference between the simulated and GRACE-based TWS anomaly are evident, especially in sub-basins with relatively smaller area and elongated shape (e.g., Purus and Japura). Note that accuracy of GRACE-model agreement is generally low in such small basins due to high bias and leakage correction errors (Chaudhari et al., 2018; Felfelani et al., 2017; Longuevergne et al., 2010), reflected by higher RMSE values in Figure 2. Simulated TWS evidently follows precipitation anomalies (shown in grey bars in Figure 2), implying that any uncertainties in the precipitation forcing could have directly impacted TWS. For example, the simulated TWS peak in 2002 in the Solimoes River basin results from the anomalous high precipitation, however this could not be validated due to a data gap in GRACE. Overall, the model performance is better in the first half of the simulation period (i.e., 2002-2008) compared to the second half, especially in the western sub-basins including the Solimoes and Japura, which could be partially attributed to the decreasing trend in the precipitation forcing noted in Figure S6.

Figure 2 also shows the seasonal cycle including the contribution of different storage components to TWS. In all the basins, simulated seasonal cycle matches extremely well with GRACE, adding more confidence to the model results. TWS signal is

sturdily modulated by the sub-surface water storage, demonstrating the importance of groundwater in the Amazon, especially in the southwestern sub-basins. The inverse relationship in the seasonal cycle of two sub-surface water stores, viz. soil moisture and groundwater, is readily discernable in Figure 2, which is caused by the competing use of the sub-surface compartment by the two terms (Felfelani et al., 2017; Pokhrel et al., 2013). However, in some sub-basins, such as the Purus, Solimoes and

Negro, the low-lying areas with large floodplains causes flood water storage to be equally prominent.

## 3.3 Trends in Simulated TWS and Comparison with GRACE

Here, we present a more detailed examination of the simulated TWS by comparing its spatial variability and trend with GRACE data. Because a shift in agreement between model and GRACE was detected in Figure 2 and S7, we conduct a trend analysis for two different time windows: 2002-2008 and 2009-2015 (Figure 3). It is evident from Figure 3 that the model captures the

general spatial pattern of TWS trend in GRACE and its north-south and east-west gradients especially for the first half of the analysis period; however, notable differences are evident in the second half (2009-2015), particularly over the Madeira River basin. This is a noteworthy observation given that the basin averaged TWS variability matches extremely well with GRACE data (Figure 2), and thus warrants further investigation. There could be a number of factors contributing to the disagreement, some of which could be model-specific (e.g., wet bias in simulated discharge; Figure S3); however, this is a general pattern

observed in many hydrological models as reported in a recent study (Scanlon et al., 2018).

Scanlon et al., (2018) indicated a low correlation between GRACE and models, which they attributed to the i) lack of surface water and groundwater storage components in most of the models, ii) uncertainty in climate forcing and iii) poor representation of human intervention in the models (Scanlon et al., 2018; Sun et al., 2019). Here, we shed more light on the disagreement issue by investigating the contributions from the explicitly simulated surface and sub-surface storage components and their

latitudinal patterns, addressing the first concern noted above which is the most critical among the three in the Amazon because of varying contribution of different stores across scales (Pokhrel et al., 2013). Figure 4 shows trends in TWS anomalies from GRACE products and the LHF simulation for the complete model-GRACE overlap period (i.e., 2002-2015) with climatology and with climatology removed; for LHF results, the surface and sub-surface component contributions to the TWS are shown. Also shown in the figure are the zonal means.

Simulated TWS from LHF model displays a higher correlation with GRACE trends compared to most of the global models discussed in Scanlon et al., (2018). Due to the incorporation of a groundwater scheme and other surface water dynamics, trend in basin-averaged TWS with climatology removed for the Amazon River basin is found to be -1.64 mm/yr, much less negative than most of the simulated TWS trends reported in Scanlon et al., (2018). The difference in the sign of trend can partly be explained by the negative trend observed in the WFDEI precipitation (Figure S6), concentrated over the Andes region which

eventually drains into the mainstem of the Amazon through the Solimoes River. Due to steep topography, the impact of decreased precipitation over the Andes range is carried over to its foothills in terms of runoff, hence corresponding well with the negative trends in simulated surface water storage over the Central Amazon (Figure 4). Lower recharge rates in the region with decreasing precipitation trend (Figure S6) are also very likely, which is supported by the negative trend visible in the sub-

surface water storage in Figure 4, over the northwest region of Amazon. Hence, it can be concluded that, even though the model shows some bias in TWS compared to GRACE data, the model accurately represents the key hydrologic processes in the Amazon basin; yet, these results should be interpreted with some caution while acknowledging the uncertainty in the forcing dataset. We also emphasize that it is important to evaluate models using spatiotemporal trends, especially with GRACE, instead of just using the basin averaged timeseries, a commonly used approach in most previous studies.

## 3.4 Interannual and Interdecadal TWS Change and Variability

Figure 5 show the interdecadal shifts in mean simulated TWS (total and its components) for the simulation period. Several observations can be made from this figure. First, the change between 2010s-2000s suggests high negative anomalies in all the water stores especially over Central Amazon. This is likely a result of increasing drought occurrence and severity in the region (e.g., the 2010 (Lewis et al., 2011; Marengo et al., 2011) and 2015 (Jiménez-Muñoz et al., 2016) Amazonian droughts). Second, although, the 2000s encompassed one of the severe Amazonian droughts viz. 2005 (Marengo et al., 2008; Zeng et al., 2008), its impact was not pronounced in terms of the decadal mean, which could be due to the offset caused by anomalous wet years including 2006 and 2009 (Chen et al., 2010; Filizola et al., 2014). Third, we find an increase in river water storage in the northwestern region and decrease in the southwest of the Amazon on a decadal scale (Figure 5, column 1, row 2), which is in line with the findings reported in previous studies based on the observed streamflow in 18 sub-basins for the 1974-2004 period (Espinoza et al., 2009; Wongchuig Correa et al., 2017).

The most remarkable feature we observe in Figure 5 is the exceptional interdecadal shifts between the 2000s and 2010s. Central and northwestern part of the Amazon region, encompassing the Negro and Solimoes, along with some parts of the Madeira in southwest, experienced a major decadal dry spell compared to the previous decades. Although a major part of this decadal dry condition could be attributed to the decreasing trend in input precipitation discussed in section 3.3 (Figure S6), the regional hydrologic changes in terms of TWS are also prominent. Another peculiar phenomenon observed at the decadal scale is the start of the negative anomaly in groundwater storage over the Central Amazon. A small but spatially well distributed below-decadal-average water table (dictated by groundwater storage) is evident in the Central Amazon region and the upper stretches of the Madeira basin during the 2010s (Figure 5, column 3, row 4). Since the water table is shallow and groundwater is the major contributor of streamflow in this region (Miguez-Macho and Fan, 2012a), some part of the negative anomaly in surface water stores can be attributed to the below-decadal-average groundwater table.

Significant long-term trends in simulated TWS and its components are evident in sizeable portions of the basin (Figure 6). While a negative trend is found in the southern and southeastern regions (e.g., Madeira, Tapajos, Xingu and Tocantins), the trend is positive in the northern and western regions (Solimoes and Negro) (see Figure S9 for basin averaged trends). Being the major contributor, sub-surface water storage mimics the trend patterns in TWS (see Section 3.2). On the contrary, surface water storage trends are mainly dominated by floodwater and are concentrated along the main stem of Amazon and the upper reaches of the Negro. The positive trends in floodwater can be explained by the corresponding trends in input precipitation (Figure S5). Excess precipitation in sub-basins, such as the Solimoes and Negro, which are characterized by a high topographic

gradient, is directly translated in the surface water storage, in this case floodwater. Although a corresponding increment in river water storage is also expected, its smaller storage makes the trend magnitudes negligible. Nominal negative trends, but significant, in floodwater storage are found in the upper reaches of Madeira as well, corresponding to the negative trends in input precipitation over that region.

To provide an in-depth understanding of the interdecadal changes occurring in the Amazon region and to determine whether the changes observed in Figure 5 are significant, we applied a t-test methodology to the long term TWS anomalies at basin and sub-basin levels. The spatial changes observed in Figure 5 are summarized with their interdecadal significance in Table S1, along with the decadal means and standard deviations. Significant change at 99% level is found in Negro River basin throughout the study period, followed by the Solimoes River basin exhibiting significant change in the last three decades.

These changes can be attributed to the corresponding changes in precipitation (Figure S5), which follow a similar change in respective basins. However, the significant hydrologic changes in the Tocantins and Madeira can be primarily attributed to LULC changes, as the corresponding changes in precipitation were relatively negligible. For example, the Tocantins River basin underwent major LULC changes in response to heavy deforestation caused by dam construction and cattle farming (Costa et al., 2003) until policies were imposed in 2004 by the Brazilian government (captured in the ESA dataset, Figure S8).

Similarly, the Madeira River basin also endured major LULC changes in the late 1990s which were dominated by agricultural expansion (Dórea and Barbosa, 2007).

### 3.5 Interannual and Interdecadal Drought Evolution

### 3.5.1 Severity of TWS-Drought

In this section, we examine the time-evolution of droughts and quantify their impacts on TWS variability by using TWS-DSI.

The use of TWS-DSI enables the depiction of a "bigger picture" encompassing all water stores that represent the vertically integrated total water availability during droughts and dictate the streamflow. Figure 7 shows the TWS-DSI for individual Amazonian sub-basins, and the 12-month standard precipitation index (SPI) (Mckee et al., 1993) calculated from the basin-averaged precipitation timeseries. As expected TWS-DSI follows a similar pattern of the SPI but differences in the index peaks can be noted for the drought years. For example, the 2005 drought was prominent in terms of TWS in the southwest region,

comprising of Purus and Madeira rivers, with TWS-DSI going as high as -3, whereas the corresponding SPI were -1.78 and -2.2, respectively. Similarly, severe TWS drought (e.g. 2001) is detected in the southeastern basins of Amazon (Madeira, Xingu and Tocantins), however, the corresponding SPIs are negligible; the sub-surface storage (major contributor of TWS in these sub-basins) characteristic can be noted in these cases which has a delayed response from the preceding series of low precipitation events due to slow residence time.

The impact of drought conditions on TWS is quantified by examining the seasonal dynamics in the simulated sub-surface water storage for the four most extreme historical drought years during the simulation period (Figure 8). Although no clear trend can be seen in terms of the evolution of the drought impact on sub-surface water storage, the spatial variability between

different drought years is readily discernible. For example, the 1995 and 2010 droughts more or less had a similar magnitude and spatial impact on the sub-surface storage, however, the 2005 drought was more intense and dramatic in the Solimoes River basin; findings also noted in previous studies (Marengo et al., 2008; Phillips et al., 2009; Zeng et al., 2008). Similarly, the more recent drought in 2015 had a more pronounced impact in the eastern and northeastern region and average impact on the other parts of the basin. Due to the shallow water table in the Amazonian lowlands, sub-surface storage acts as a buffer during the low precipitation events, hence facing higher anomalies during drought conditions compared to the long-term mean. As the Negro river (i.e. Northern region of Amazon) basin experiences an opposite seasonal phase compared to rest of the Amazon region, the drought conditions in this basin are observed during the period of December to March. The opposite seasonal cycle of precipitation and flooding in the north and south banks of the Amazon mitigates the amount of flood and droughts in the basin as a whole, while resulting in more dramatic flood or drought in particular sub-basins (e.g. Tocantins, Tapajos and Madeira).

### 3.5.2 Time Evolution of Dry Season Total Deficit and TWS Release

The dry season TWS variability is examined by using the cumulative difference between PET and P, termed as the TWD (see Section 2.7). Further, to examine the response from TWS against TWD, we quantify the TWS-R, hence creating a supply-demand relationship between them. Figure S10 shows TWD, the corresponding TWS-R, and the total contribution of the surface water storage to TWS-R for the extreme drought years during 1980-2015 compared to their respective long-term means. Spatial patterns in TWD and TWS-R are analogous to the patterns in the simulated sub-surface storage during the months of September to November (SON) as seen in Figure 8. We find that TWS-R receives a fairly equal contribution from surface (along the rivers) and sub-surface (soil moisture and groundwater) water stores (rest of the region); however, the latter is more dominant during drought years. A clear positive trend in drought years is visible in Figure S10, indicating an increase in TWS-R, with significant sub-surface contribution, especially in the southeastern part of Amazon. This change can be directly attributed to the major LULC changes occurring in the basin, causing loss of TWS to evapotranspiration through agricultural expansion, especially in the Tocantins, Xingu, Tapajos and Madeira river basins (Chen et al., 2015; Costa et al., 2003; Dórea and Barbosa, 2007).

### 3.5.3 Hydrological drought trends in Amazonian sub-catchments

The hydrological drought behavior of each sub-basin is characterized by quantifying the drought days per year at the Level-5 Hydro-Basins scale (Lehner and Grill, 2013), referred here to as 'sub-catchments'. Based on the streamflow simulated at the most downstream grid in the sub-catchments, temporal trends for the 1980-2015 period are calculated and presented in Figure 9. Significant trends in drought durations are discernible in the Tapajos and Madeira sub-basin along with the southeastern portions of the Amazon, congruent to the heavy deforestation activities found in these sub-basins (Chen et al., 2015; Costa et al., 2003; Dórea and Barbosa, 2007). Although, LULC changes, such as deforestation activities, generally increase streamflow and are also known to offset the impact on streamflow caused by decrease in precipitation over the Amazon (Panday et al.,

2015), this mechanism is dominant mostly during the wet season. In the dry season, however, the streams in the Amazon are fed primarily by the sub-surface water storage (see Section 3.2), which is negatively impacted by deforestation activities (e.g., increased regional evapotranspiration).

## 3.6 Comprehensive Characterization of Amazonian Droughts

As a first attempt to comprehensively characterize the Amazonian droughts, we present a summary of all the drought characteristics discussed in the previous sections on a spider plot (Figure 10). Each spider plot is a representation of a drought year with respect to the i) causes of drought and their type in terms of common indices, ii) response of different water stores, such as TWS, to the drought event, iii) role of groundwater storage in alleviating the dry conditions on surface, and iv) the spatial impact of the drought in different sub-basins of the Amazon. Although no significant trend in the combined drought
characteristic is apparent, figure 10 provides important insights on the variability of Amazon droughts. It is evident from the figure that the drought variability over the years was significant in terms of both magnitude and spatial impact. The most notable feature in Figure 10 is the distinct relationship between SPI and drought duration. For example, during the 1995 drought, most of the river basins (e.g., Tocantins, Tapajos, Xingu, and Negro) experienced significant meteorological and TWS droughts, however, the severity of hydrological droughts was relatively negligible in those basins. Groundwater-surface
water exchange is the key mechanism behind this unique behavior, causing groundwater to fulfill the drought deficit in streamflow over the basin. Due to shallow water tables at the downstream end of these basins, significant quantity of groundwater is fed to the rivers, which manifests as high peaks in total groundwater release evident in Figure 10. Similarly, high number of drought days are found corresponding to less groundwater release, such as during the 1995 drought in Madeira. On the contrary, TWS-DSI, generally follows the same pattern as that of SPI but with a lesser magnitude, which can be
attributed to the delayed response from groundwater.

Further, the behavior of the Amazonian sub-basins can be characterized by the shape of the polygon formed by the comparison of different aspects of past droughts. The convex and concave characteristic in the plots mainly depends on the interrelation between meteorological and hydrological drought indices, which is further controlled by the sub-surface water storage. A convex polygon indicates lower groundwater contribution to streamflow in the sub-basin, such as in Purus during 1995 and
2005, whereas a concave polygon suggests higher groundwater release to streamflow in that particular year.

## 3.7 Intensification of the Amazonian Dry Season

Results suggest an increasing trend in TWD with significant decadal variability over the Amazon and its sub-basins, indicating an increase in dry season length over the past 36 years (Figure 11). Further, the increasing gap between TWD and TWS-R suggest an intensified terrestrial hydrologic system over the dry season during the study period. As the LULC impact is partly
accounted for in the PET calculations (i.e., through changing surface albedo), the river basins with substantial LULC change, such as Madeira, Tapajos, Tocantins and Xingu, portray higher TWD trend magnitudes (significance > 95%). The peaks in the TWD corresponds well with drought years, for example, the peaks in the TWD for Madeira are analogous to the drought years

(e.g., 1988, 1995, 2005 and 2010). Due to this definitive response to drought conditions, TWD is also used to characterize historical drought events in the earlier sections. We note that the trends in the total deficit should be interpreted with caution as the uncertainty in the forcing could have affected TWD and TWS-R trend estimates.

We find that the river basins housing high altitudinal areas (Purus, Solimoes and Negro) have a fairly balanced relationship between TWD and TWS-R, but southern and southeastern sub-basins exhibit a higher water deficiency (Figure S11) with approximately 2 to 3-fold differences between TWD and TWS-R during regular years. For drought years, however, the difference between TWD and TWS-R is even higher, creating highly anomalous dry conditions in the sub-basins. Consistent higher values of TWD in southern and southeastern sub-basins of Amazon further highlights the intensification of the dry season with increasing water deficiency corresponding to an almost constant water supply from TWS-R. This phenomenon is also highlighted in Espinoza et al., (2016), which showed an significant increase in dry day frequency in the central and southern parts of Amazon. Results from this study combined with the reported increasing trend in wet season (Gloor et al., 2013), implies an overall intensification of the Amazonian hydrological cycle.

## 4. Conclusion

In this study, we examine the interannual and interdecadal trends and variability in the terrestrial hydrological system in the Amazon basin and its sub-basins, with a focus on droughts and their time evolution during the 1980-2015 period by using a continental-scale hydrological model Leaf-Hydro-Flood (LHF) and terrestrial water storage (TWS) data from GRACE satellite mission. For the first time, we provide a comprehensive characterization of extreme drought events in the Amazon basin during the past four decades, while categorizing them with respect to their i) cause, ii) type, iii) spatial extent, and, iv) impacts on different water stores. We also provide an in-depth understanding of the interrelation between different drought types and the corresponding response of the sub-surface storage to surface drought conditions. Our key findings are summarized below.

First, the LHF model simulates the basin averaged TWS variations and seasonal cycle remarkably well for most of the sub-basins compared to GRACE data, however, some differences are observed in the spatial distribution of temporal trends for post-2008 period. We find that this discrepancy is caused primarily by the uncertainty in surface water storage simulations along the mainstem of the Negro and Amazon, whereas uncertainty in sub-surface storage prevails over the Andes. Second, the 2010-2015 period was found to be the driest in the past four decades due to an increase in frequency and severity of droughts. A t-test conducted on the TWS timeseries also indicated significant changes at the 99% level in the decadal mean TWS in the Negro and Solimoes sub-basins. Third, high negative long-term trends in TWS and increasing divergence between dry season total water deficit (TWD) and corresponding TWS release (TWS-R) indicate significant drying in sub-basins such as Madeira, Tapajos, Xingu, and Tocantins. Basin-averaged trends indicate that the Amazon is getting wetter (1.13 mm/yr), however, its southern and southeastern portions are getting drier. TWD is also found to be higher than TWS-R in these sub-basins, with approximately a three-fold difference between the two during some drought years, indicating a strengthening dry season in the region. Fourth, most of the extreme meteorological droughts do not propagate to hydrological droughts

significantly, as the deficit is absorbed by the subsurface water storage and further reducing TWS drought severity compared to that of a meteorological drought in the Amazonian sub-basins.

Altogether, these results provide important insights on the interannual and interdecadal hydrological changes and the key mechanisms that govern drought events in the Amazon, along with a novel way of categorizing basin behavior during drought occurrence (Figure 10). This framework can be applied to better predict the future hydrological conditions and their corresponding socio-economic impacts toward taking measures to mitigate the drought impacts and facilitate a relatively facile transition of the local population through a future drought event. Basin drying trends reported in this study can also provide key leverage by applying them toward anticipation of the future hydrological conditions for sustainable management of water resources. We also highlight the importance of using spatiotemporal trend estimates for model validation, especially with GRACE, instead of the commonly employed approach of timeseries comparison. Improvement in the correlation between the temporal trends in simulated TWS and GRACE anomaly through the inclusion of a prognostic groundwater scheme which allows dynamic groundwater-surface water interactions in the model framework is also highlighted. Further, the need to investigate the effects of uncertainties in model forcing to TWS simulations is noted because we find that the trends in precipitation are strongly propagated to TWS simulations.

A limitation of the present study is that the effects of irrigation and manmade reservoirs are not yet incorporated in the model. The basin-wide effects of the existing dams in the Amazon are small (Pokhrel et al., 2012a); however, as more dams are added across the basin, it will become critical to account for such effects. Model improvement is underway (Pokhrel et al., 2018; Shin et al., 2018), and these issues will be addressed in our forthcoming publications. Despite some limitations, this study significantly advances the understanding of changing Amazonian hydrology, and our results have important implications for predicting and monitoring extreme droughts in the region; the research framework can also be applied to other global regions undergoing similar hydrological changes.

**Author Contribution**

SC, YP, EM designed the research; YP, SC, and GM setup the model; SC performed simulations, analyzed results and prepared the draft; all authors discussed the results and wrote the manuscript.

**Competing interests**

The authors declare that they have no conflict of interest.

## Acknowledgement

This study was supported by funding from a National Science Foundation award (#1639115). River discharge data are obtained from the Agência Nacional de Águas (ANA) in Brazil website (http:// hidroweb.ana.gov.br). We obtained the annual land cover maps from European Space Agency Climate Change Initiative Land Cover project (http://maps.elie.ucl.ac.be/CCI/viewer/index.php). Simulations were conducted at Cheyenne (doi:10.5065/D6RX99HX) provided by NCAR's Computational and Information Systems Laboratory sponsored by the National Science Foundation. None of these funding sources or agencies should be held responsible for the views herein. They are the sole responsibility of the authors.

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

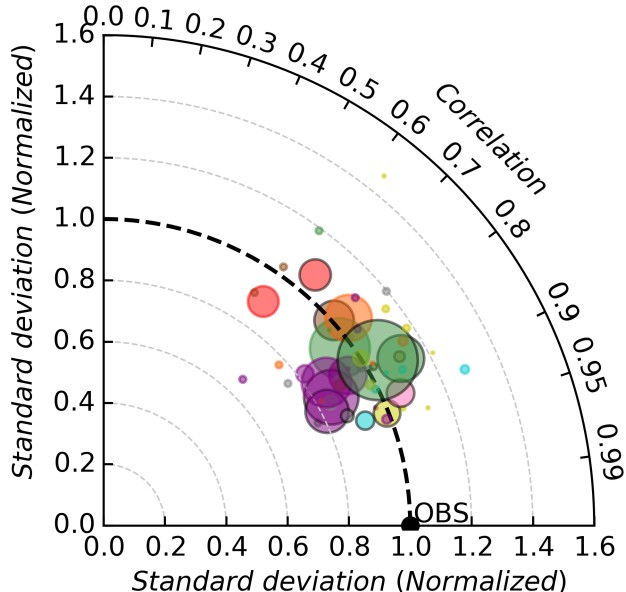

**Figure 1 – Taylor diagram showing the correlation and standard deviation ratio between the simulated and observed streamflow at 55-gauge stations across the Amazon. The locations of the 55-gauge stations are shown in Figure S2. Highlighted points with black border are the gauge stations for which timeseries comparisons are shown in Figure S3 and S4. Size of the markers indicates the annual mean simulated streamflow at that station whereas the color indicates the Amazon sub-basin in which the station is located. The linear distance between each marker and the observed data (i.e., OBS; the black dot) is proportional to the root mean square error (RMSE).**

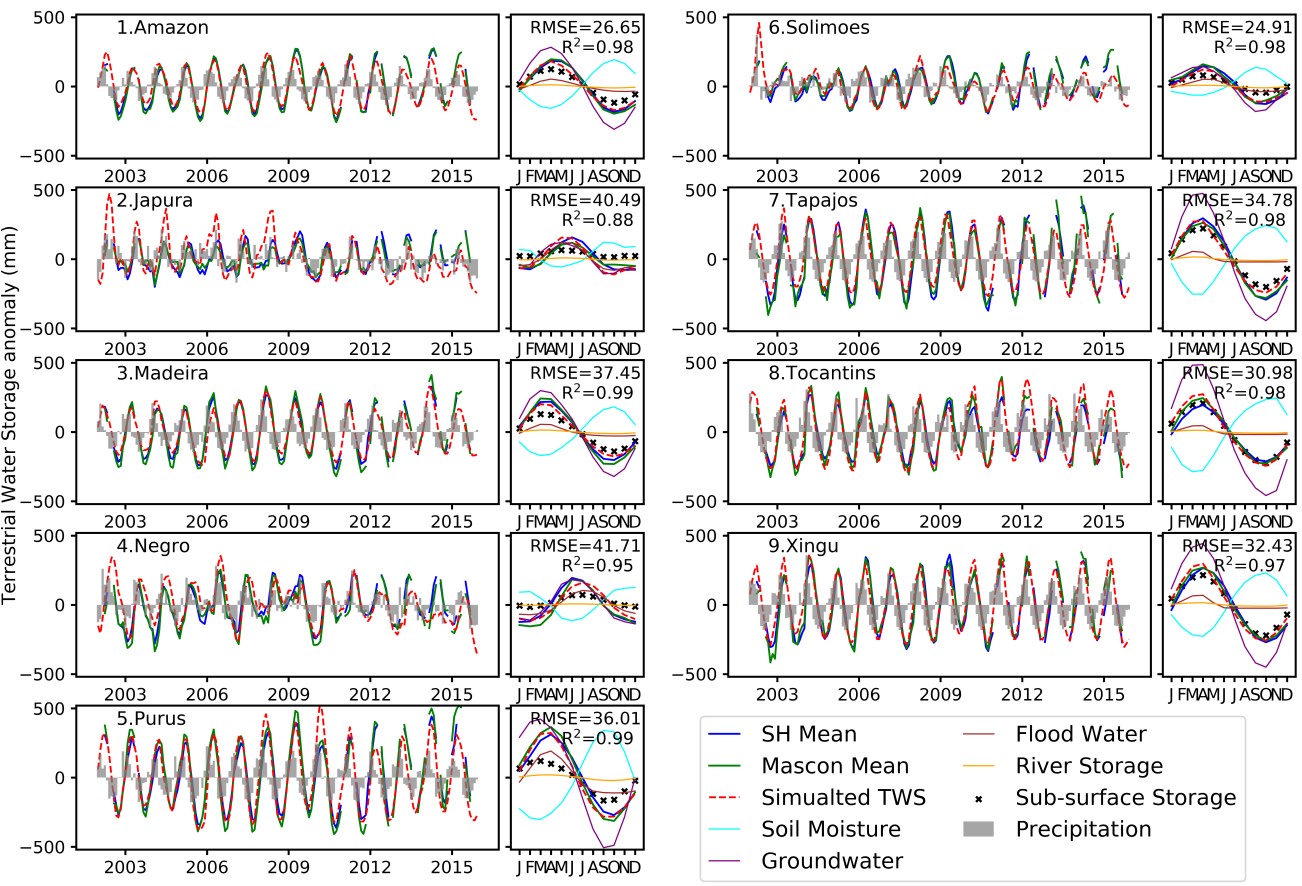

**Figure 2 – Comparison of simulated TWS anomalies from LHF and TWS anomalies obtained from GRACE for the entire Amazon and its eight sub-basins for 2002–2015 period. Basin averaged precipitation anomalies obtained from WFDEI forcing dataset are also shown as grey bars. Seasonal cycles of GRACE and simulated TWS are shown in the right panel of each basin along with the simulated individual TWS components. GRACE results are shown as the mean of the spherical harmonics solutions from three different processing centers (i.e., CSR, JPL, and GFZ) and mascon solutions from CSR and JPL. Simulated TWS anomalies are calculated with respect to the GRACE anomaly window of 2004-2009 for consistency.**

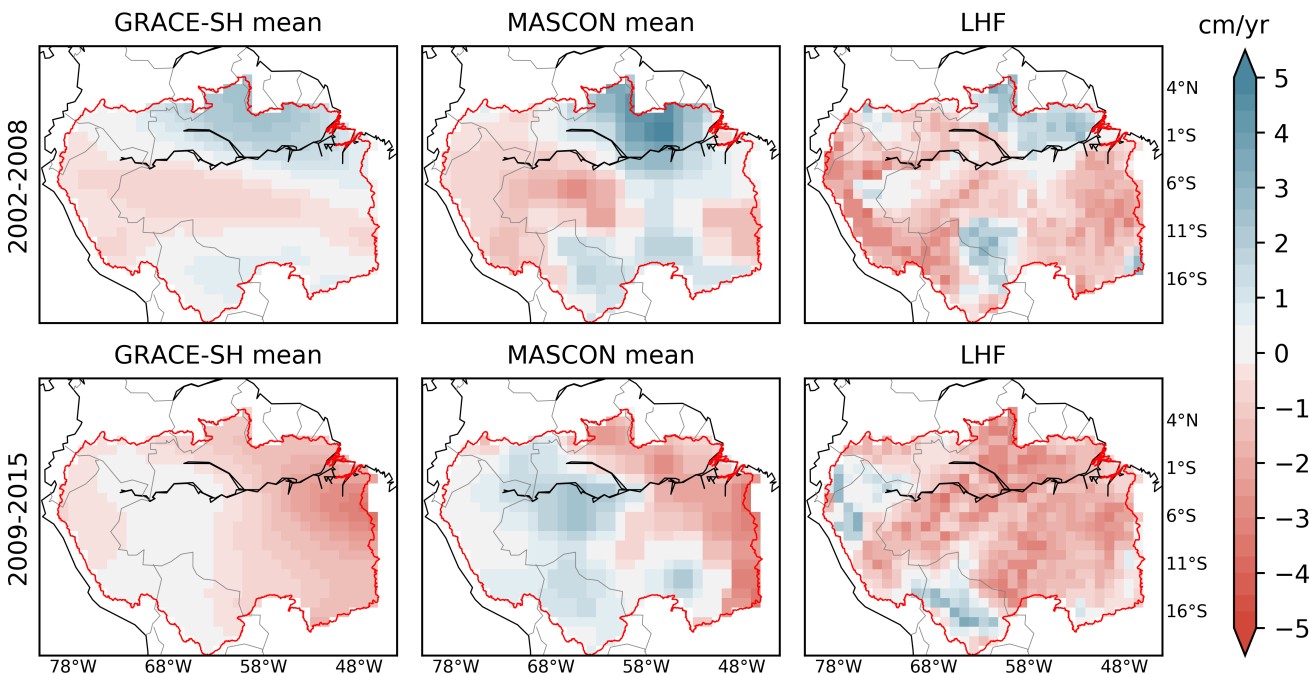

**Figure 3 − Temporal trend of GRACE solutions compared to the trend in simulated TWS from LHF for the Amazon River basin for two different time periods. GRACE-SH trend displayed, are mean trends computed from water thickness anomalies obtained from CSR, GFZ and JPL processing centers, whereas the mascon mean trend is computed from anomalies obtained from CSR and JPL centers.**

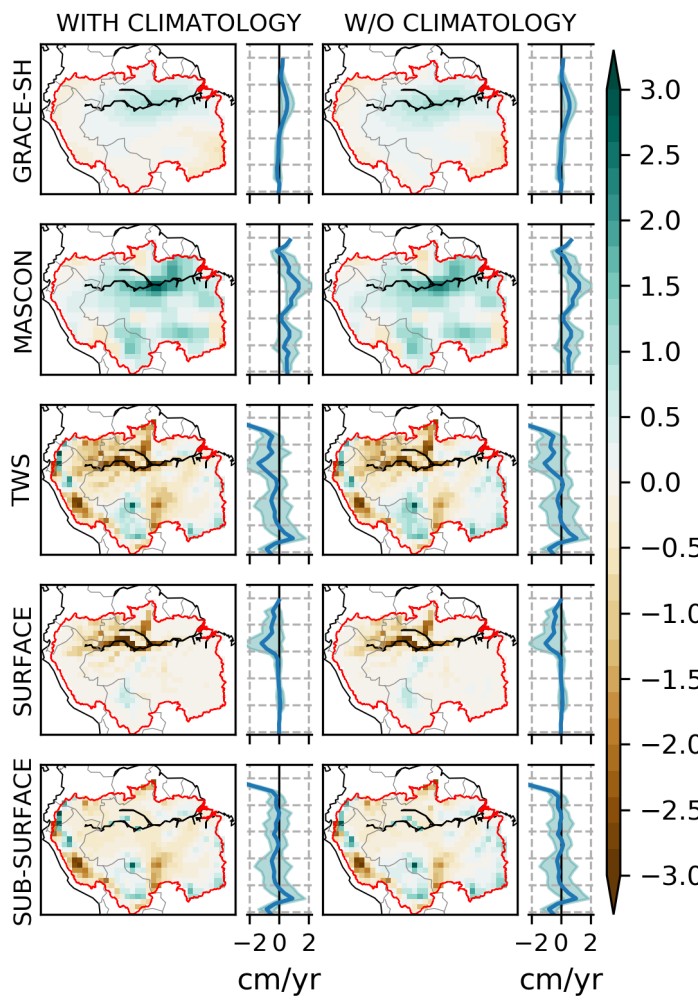

**Figure 4 – Same as in Figure 3 but for the complete model-GRACE overlap period (i.e., 2002-2015). The latitudinal mean is shown on the right side of each panel.**

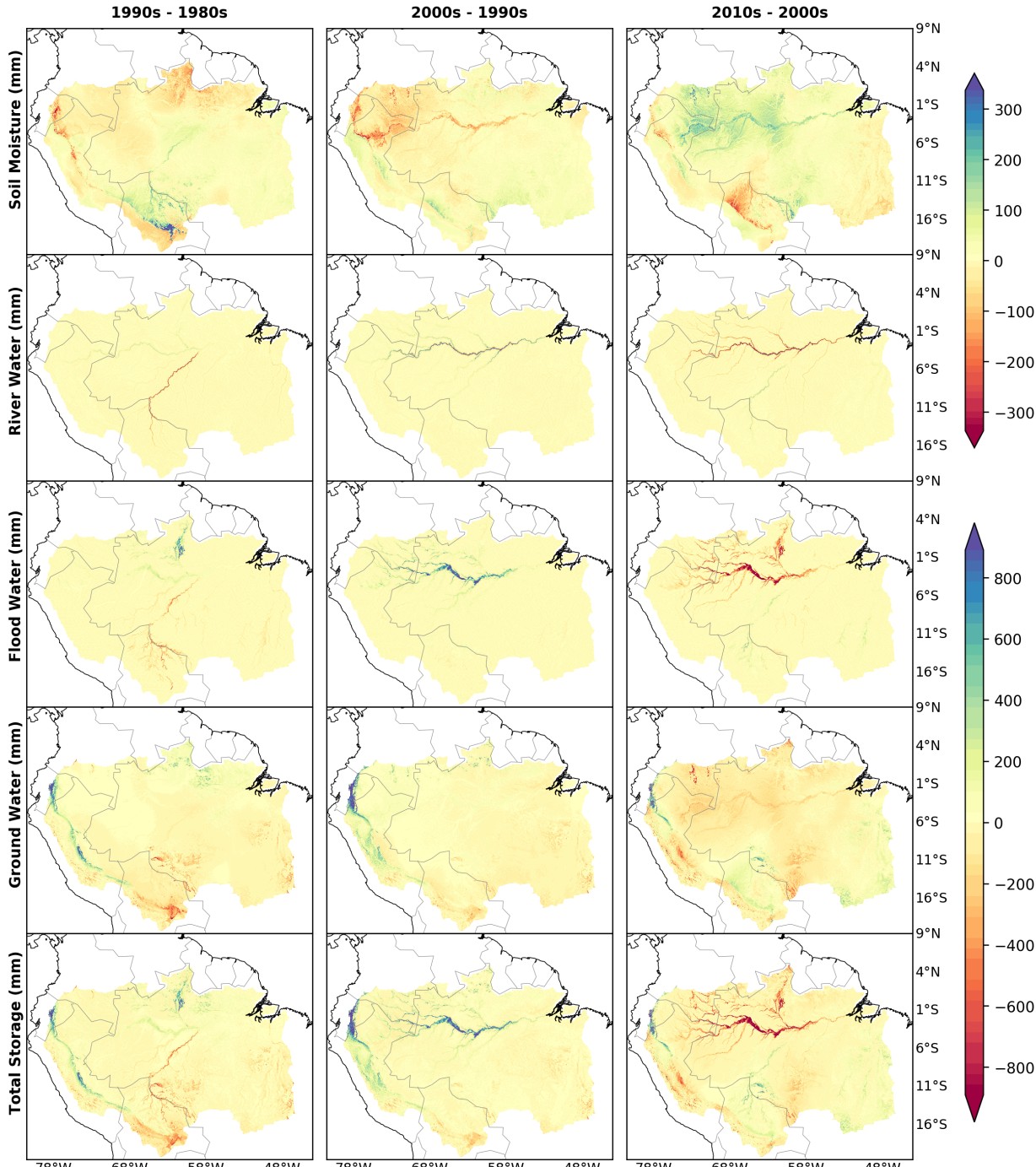

**Figure 5 – Interdecadal difference between individual water store and TWS storage for the period of 1980-2015 at the original ~2 km model grids. The changes are displayed as the difference between consecutive decadal means for TWS and its components. Decadal windows are: 1980-1989 as 1980s, 1990-1999 as 1990s, 2000-2009 as 2000s and 2010-2015 as 2010s. Note that the 2010s period consists of only six years and the ranges of color bars differ among the plots.**

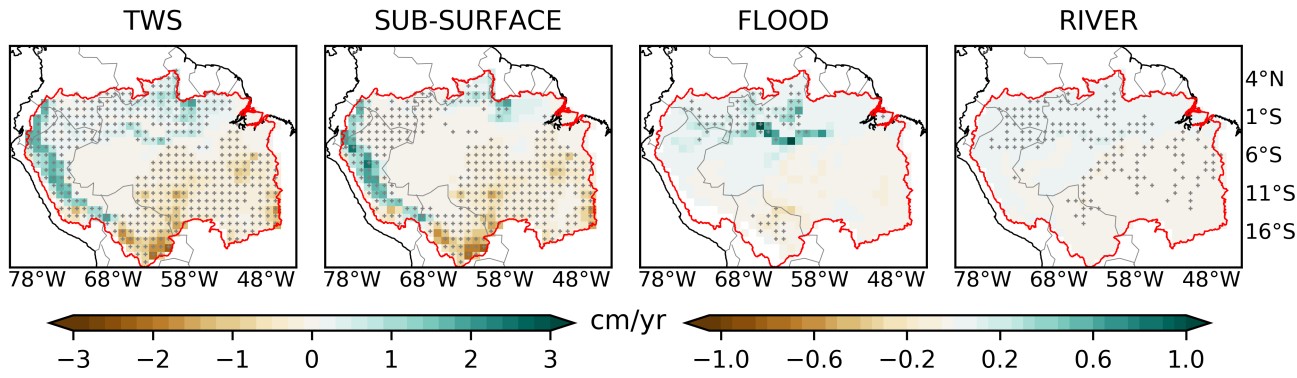

**Figure 6 – Temporal trend in simulated TWS and its components (i.e., sub-surface water, flood water, and river water stores) for the period of 1980 to 2015 expressed in cm/yr. Markers indicate significant trends at 99% level. Note that the ranges of color bars differ among the plots.**

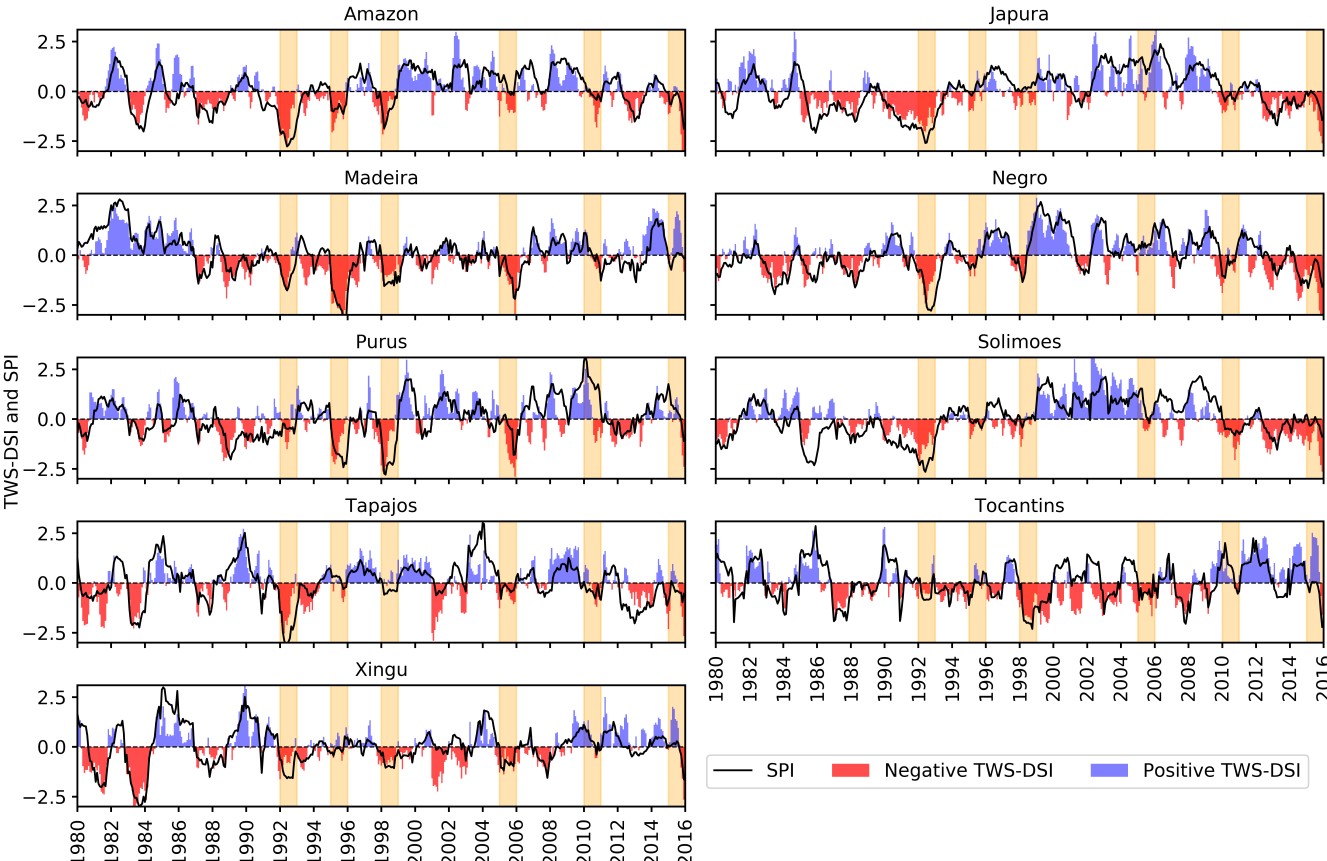

**Figure 7 – TWS drought severity index (TWS-DSI) calculated using the simulated TWS from LHF for Amazon and its sub-basins. TWS-DSI are calculated using basin averaged TWS anomalies on a monthly scale. Shaded areas indicate the severe drought years reported in the past literature. Black line is the 12-month standardized precipitation index (SPI) calculated by using basin-averaged precipitation data from WFDEI forcing dataset.**

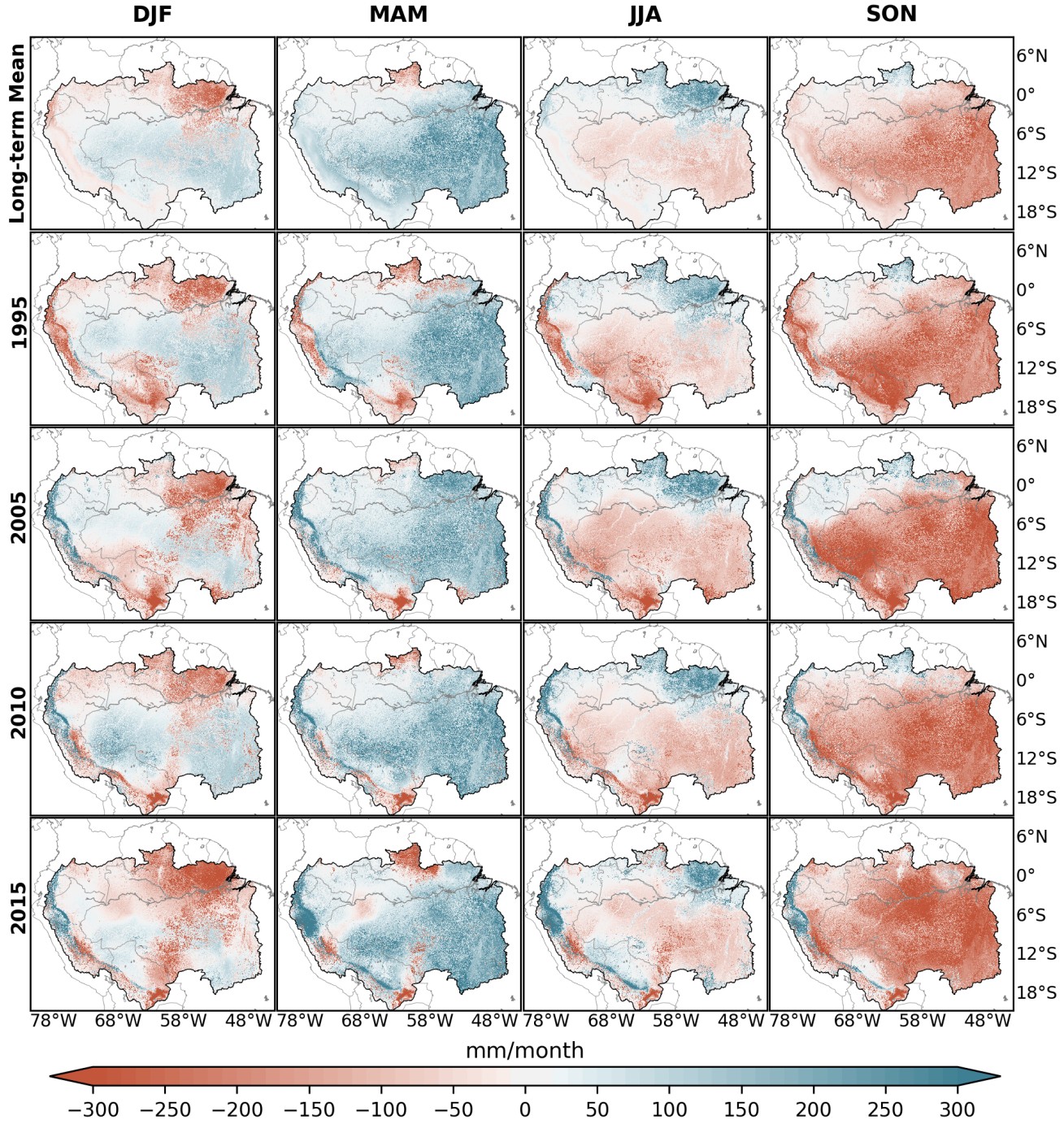

**Figure 8 – Seasonal dynamics of simulated subsurface water storage from LHF in the Amazon River basin for extreme droughts during the simulation period. Long term mean is the mean seasonal anomaly for the 1980-2015 period, where DJF is December to February, MAM is March to May, JJA is June to August, and SON is September to November.**

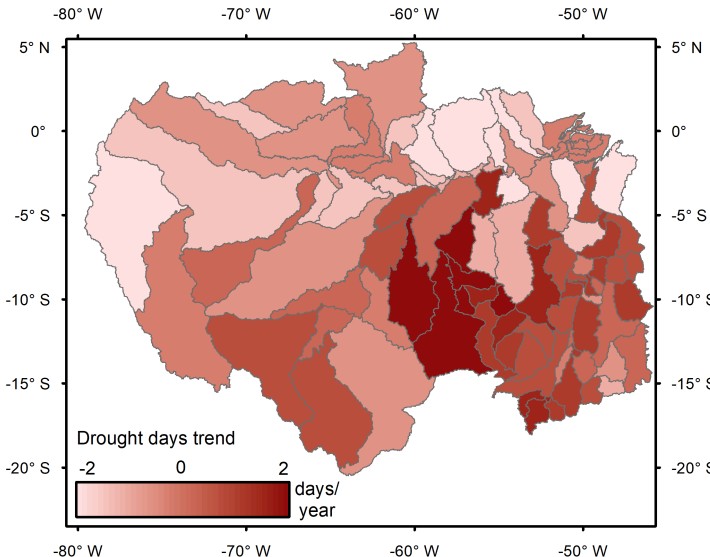

**Figure 9 – Trends in drought duration per year in the Amazon at a Level-5 Hydro-Basins scale as defined in Lehner and Grill, (2013), derived by using the Q$_{90}$ threshold from the simulated streamflow by the LHF model. Darker colors indicate the higher positive trend magnitudes.**

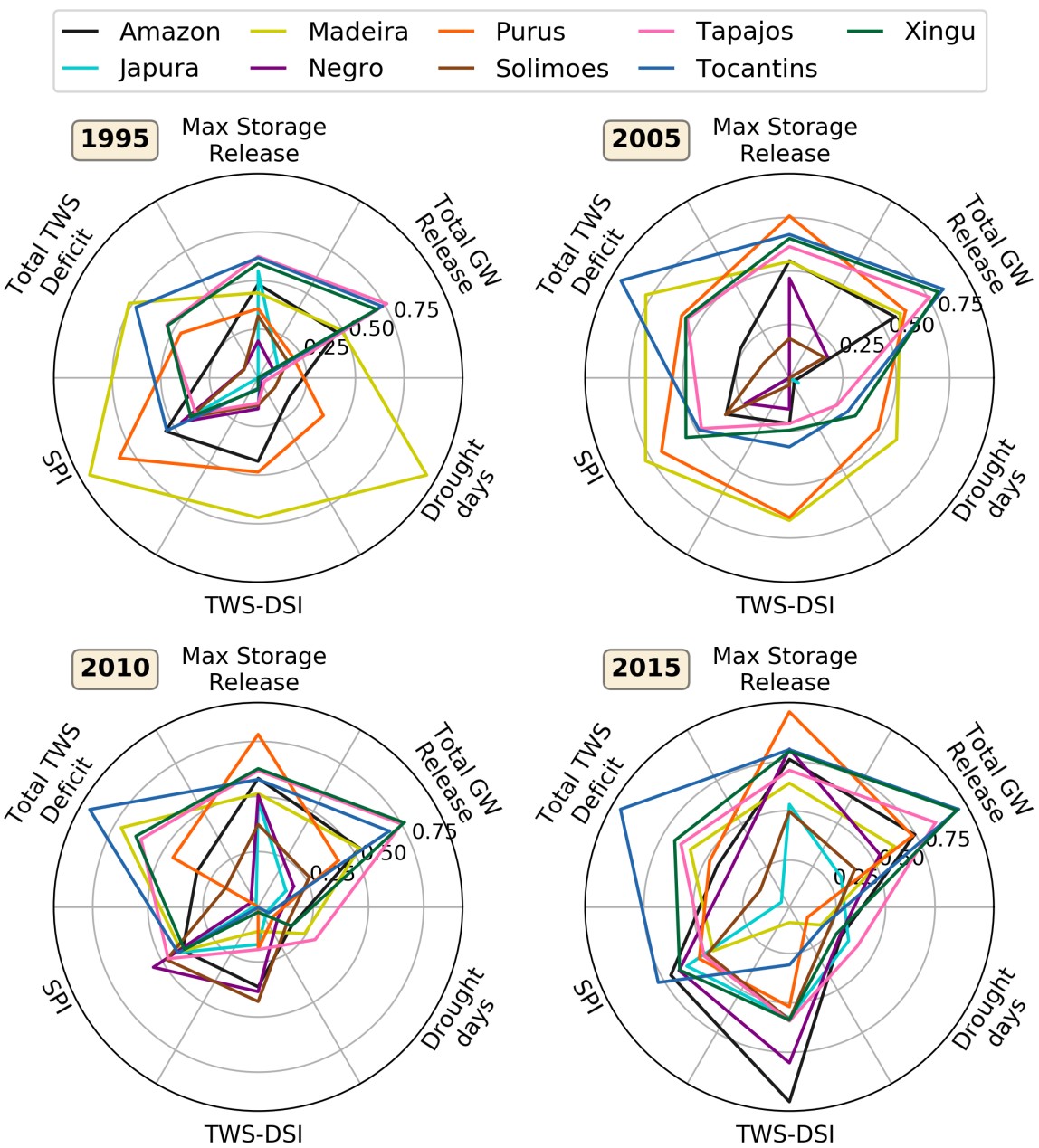

**Figure 10 – Inter-comparison and comprehensive characterization of the severe drought events during the modelling period in the Amazon River basin and its sub-basins. Color coding in each subplot represents individual river basin. Note that all variables are basin averages normalized (0-1) for each variable over all drought years. Bottom half of the variables in the figure are drought indices representing different types of droughts: TWS-DSI denotes TWS-drought severity index (Section 2.7), SPI (Standard Precipitation Index) represents meteorological drought severity, and "drought days" represents hydrological drought severity in the basin (Section 2.6). Top half of the variables quantify the water deficit in terms of total TWS deficit (cumulative PET-P), water supply as the TWS release (Max storage release) and the groundwater contribution of TWS release (Total GW Release).**

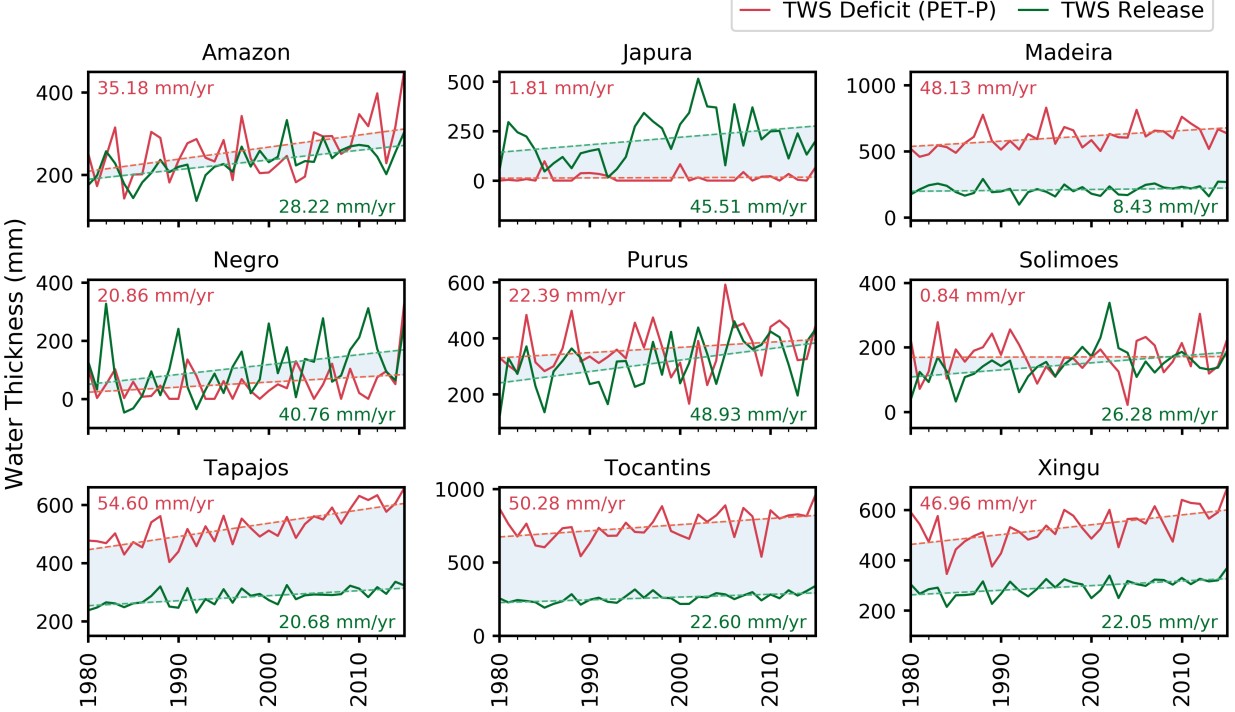

**Figure 11 – Trends in dry season total deficit (TWD) quantified as the cumulative difference between potential evapotranspiration and precipitation (PET-P) and corresponding simulated TWS release (TWS-R) from LHF for Amazon and its sub-basins.**