# Peer review of "Multi-decadal Hydrologic Change and Variability in the Amazon River Basin: Understanding Terrestrial Water Storage Variations and Drought Characteristics"

_Hydrology and Earth System Sciences, 2019_

## Referee Comment (RC1) · Anonymous Referee #1 · 3 Apr 2019

Dear authors,

I want to congratulate for a very well written, thought-through, and structured manuscript covering a wide range of topics and providing a very extensive analysis. Applying a continental model to assess water storage variations can be a key contribution for a better understanding of future threads to endangered ecosystems such as the Amazon River basin.

Before acceptance is possible, however, I have a few critical comments which are provided hereafter separated as general and specific comments on the submitted work.

**General comments**

- The work contains a lot of modelling work with subsequent extensive validation and comparison of model results with a range of observations. In this sense, it presents a "classical" technical modelling study with potential to be scaled up to other basins. Since droughts can have pronounced societal impact, I would like to read more about how the presented work can help to not only "anticipate future hydrological conditions" but also how this knowledge could be used as leverage to tackle present and future challenges of water management in the Amazon. Both in the Introduction and in the Conclusion adding the societal dimension and possible added value of your work would be of great benefit to the manuscript.

- Throughout the manuscript, there are quite some adverbs such as "relatively", "extremely" and so on. Please ensure that you use those words only when absolutely appropriate.

- I found quite some instances where you describe the figures in the actual text (eg. The first seven lines of chapter 4.1). To improve readability and shorten the text, I advise to limit the descriptions to the figure captions and only refer to the figures in the text.

- You compare interannual and interdecadal results. While I do see the added value of analyzing interannual variation, I am wondering why you decided to compare decades as well? In my opinion this time interval is just not long enough to assess long-term changes. Why not assessing long-term trends over the entire climatology instead?

**Specific comments**

- Page 2/ line 17: Is it possible to associate the climate and human-induced changes more exactly with the results? What are the driving factors, is it rather the change in climate or the increased human activities that alter the (hydrologic) system in the Amazon?
- Page 2 / lines 23-26: This sentence is in my opinion a repetition of the information presented in the paragraph before (eg. With respect to streamflow reduction). It would be worthwhile considering removing any repetitive statements here and throughout the remaining manuscript.
- Page 4/ line 6: From the manuscript it does not become clear whether the "opposite trend" between model output and GRACE is only a thing in the Amazon River basin or whether it is issue also on global scale. Please provide this information so readers can get a better idea of the severity of this problem. Also, why is there no explanation available? Maybe provide a brief sentence (based on Scanlon 2018?) since it's bit unsatisfying to read at the moment.
- Page 5/ line 5 "The LHF model":
  - Even though the paper was already published, adding a flow chart would help the reader to better understand the LHF model and the modelling steps required.
  - What is the temporal resolution of model output? Please add.
  - Was the model calibrated? If so, how and using which data and parameters? If not, why not? Please add this information.
- Page 5/ line 19: What was the reasoning to use the 2 km version of LHF and not the probably faster 5 km version?
- Page 5/ line 26: It is unclear to me why you perform an extensive spin-up of 150 years but then discard the first year of the modelling period too? Were there issues with model stability or initial conditions? Please explain your choices clearly to avoid any misunderstanding.
- Page 6/ line 1 "Atmospheric forcing":
  - Why did you decide to use the WATCH data as model forcing? Why not the more recent ERA-5 data?
  - The accuracy of forcing plays an important role, also in the Amazon. The work by Towner et al. (2019, HESSD; https://doi.org/10.5194/hess-2019-44) compares several forcing data sets (ERA-I, ERA-5, and re-forecasts) with respect to their relative impact of model accuracy for the Amazon basin. I think that this could provide a good starting point for a brief discussion about the role of forcing in modelling studies and to explain your decision to use WATCH data.
- Page 6/ line 10: I understand you use annual input for the land cover? Are you confident that this is acceptable given the model runs at a different temporal resolution I assume? Please add a brief statement why you are convinced your choice made is appropriate.
- Page 6/ line 19: how did you decide to use LAI value 5 as threshold for transitioning into forest? Is this based on expert knowledge, scientific literature or just an arbitrary decision? I can image that in vegetation-rich areas (ie. With generally high LAI values) such as the Amazon this threshold can have a marked influence on the total area eventually specified as forest.
- Page 7/ line 11: did you use monthly averages? Please add this information.
- Page 7/ line 14: Do all stations cover the same period or not? And what about missing values in the time series – are there any and if so how did you treat them?
- Page 9/ line 29: Why is it that GRACE shows little agreement in small basins? Please add a brief explanation. Also, I am wondering whether using GRACE is then the right approach for

those basins – would it not be an alternative to skip the GRACE comparison for small basins where it's known a priori that agreement will be small?

- Page 11/ line 7: I have my doubts whether the manuscript presents a "state-of-the-art" framework. After all, the model used is already available for quite a while and the main novelty of the presented work is the extensive analysis with GRACE and streamflow data for the Amazon and its sub-basins for a long period of time (which is an important contribution to current process understanding!). Another example would be the WATCH forcing which could be updated with more recent data sets. If you're convince the framework is nevertheless state-of-the-art, I would like to see an elaboration in the model description section why that is the case.

- Page 14/ line 3: What was your motivation to employ HydroSHEDS basins for this specific analysis? Do these "sub-catchments" match the geographical extent of the sub-basins you are referring to in the remainder of the manuscript? If not, this choice somehow complicates the analysis by introducing another geographical unit; in this case, I would advise to stick to the sub-basin definition used for the other analyses.

- Page 15/ line 3: You are mentioning "important insights" but I don't see any further elaboration what those insights could be. Please append this information!

- Page 15/ line 4 "Intensification of the Amazonian Dry Season": This chapter could profit from discussing findings from other literature to put your results into perspective. Please add where applicable!

---

## Referee Comment (RC2) · Anonymous Referee #2 · 26 May 2019

This study applied a physics-based hydrological model and GRACE product to investigate the hydrological changes in the Amazon basin, especially the water storage and how it related to droughts, during 36 years period. The results of this study are comprehensive and the findings are significant, which improve the understanding of hydrology in Amazon. But there are still some concerns in the manuscript need to be addressed. The first two questions regard the modeling approaches. Firstly, it was mentioned that the atmospheric forcing data are spatially interpolated using a bilinear interpolation method to the model grid. The issue is, for example, rainfall events are usually local

and spatially discontinuous, whether the bilinear interpolation is appropriate for some of the climatology data. Secondly, regarding the LULC change applied to the model, LAI higher than 5 are considered as forest canopy. Then the question is, how does this approach deal with the seasonal variation of LAI as for LULC change? The manuscript consists of 5 parts, but the model descriptions in Section 2 should belong to Section 3, methods. Thus, it would be better to re-organize the contents and the structure of the manuscript. In addition, Figures S3, S6, and S8 are not referred nor discussed in the manuscript. Moreover, there are also some specific comments as below.

1. P3L13∼15, some of these 'more recent' literature are still more than 10 years old. The author should cite some real more recent papers. 2. P8L27∼29, the description of the symbols in the figure should also be presented in the figure caption. 3. P9L7, this conclusion is not easy to clarify from the figures. Please describe more clearly and specifically. 4. P9L14, the discrepancies in some basins cannot be seen from Figure S2, for example, by which metrics? 5. P12L13, the method of t-test should be described in the methodology section unless it is an ordinary t-test. 6. P14L14, it should be 'Figure 10'. 7. Figure 5, the color change of the rivers is not clear. The line widths of the rivers should be increased. 8. Figure S1 lacks the north arrow and the scale. Moreover, the author should mark all sub-basins and major rivers in this figure. 9. Figure 7, y-axis label is missing. 10. It would be better to include geo-coordinates for all spatial plots, e.g., Figure 3, 4, 5, 6, 8, 9, S4, S5, S7, and S9.

---

## Author Comment (AC1) · 4 Jun 2019

Dear authors, I want to congratulate for a very well written, thought-through, and structured manuscript covering a wide range of topics and providing a very extensive analysis. Applying a continental model to assess water storage variations can be a key contribution for a better understanding of future threads to endangered ecosystems such as the Amazon River basin. Before acceptance is possible, however, I have a few critical comments which are provided hereafter separated as general and specific comments on the submitted work. Response: Thank you for your positive evaluation of the

manuscript and the constructive comments that helped tremendously in improving the quality of the paper. We provide a detailed point-by-point response to all comments in the following. We note that we have addressed all your comments and have made necessary changes in the revised manuscript. We have also provided detailed responses to each comment to avoid any confusion along with references directed towards the revised manuscript. Please note that following the other reviewer's comment, we have made slight changes in the paper outline of the revised manuscript.

General comments:

GC1) The work contains a lot of modelling work with subsequent extensive validation and comparison of model results with a range of observations. In this sense, it presents a "classical" technical modelling study with potential to be scaled up to other basins. Since droughts can have pronounced societal impact, I would like to read more about how the presented work can help to not only "anticipate future hydrological conditions" but also how this knowledge could be used as leverage to tackle present and future challenges of water management in the Amazon. Both in the Introduction and in the Conclusion adding the societal dimension and possible added value of your work would be of great benefit to the manuscript.

Response: Thank you for the thoughtful comment. We agree that the outlook based on societal aspects will greatly benefit the manuscript, hence giving a more complete picture of the Amazonian droughts. We have provided additional discussion regarding droughts with respect to the societal dimension both in the Introduction and Conclusion. However, we have attempted to keep the discussion on the societal dimension short as this issue is not completely within the overall scope of the present study. To summarize, we discuss the broad impact of droughts on the livelihood of the local population through the disruption in fish yield, navigation, drinking water supply, etc. We also present an overview of the impact on the financial condition of the riverine population caused by incessant droughts. Further, we have provided a discussion on the application of the results presented in this study towards prediction and mitigation of

future drought conditions in the Conclusion section of the revised manuscript.

GC2) Throughout the manuscript, there are quite some adverbs such as "relatively", "extremely" and so on. Please ensure that you use those words only when absolutely appropriate.

Response: Thank you for the suggestion. We have revised the entire manuscript to avoid the use of adverbs. However, in some locations the adverbs are retained to maintain the statement's inference and to keep them concise, avoiding excessive increase in words because the manuscript is already a little long.

GC3) I found quite some instances where you describe the figures in the actual text (eg. The first seven lines of chapter 4.1). To improve readability and shorten the text, I advise to limit the descriptions to the figure captions and only refer to the figures in the text.

Response: Thank you for the suggestion. We have limited the figure descriptions to figure captions (for example, Figure 1 and 10). General changes to the text are also made to ensure a smooth flow between text and paragraphs.

GC4) You compare interannual and interdecadal results. While I do see the added value of analyzing interannual variation, I am wondering why you decided to compare decades as well? In my opinion this time interval is just not long enough to assess long-term changes. Why not assessing long-term trends over the entire climatology instead?

Response: We agree that the time interval of this study is not long enough for a complete interdecadal analysis. However, given that we simulated the Amazon hydrology for 36 years, it was worthwhile to take a first look at the hydrological changes occurring at a decadal timescale. Realizing the shorter time frame for interdecadal trend analysis, we kept the discussion limited to decadal differences rather than specific trends (section 3.4). Our simulation period also encompasses several ENSO episodes, so it is

worthwhile utilizing the simulations to examine how the hydrology is changing decade by decade. Moreover, studies have shown that precipitation exhibit opposite trends over the northern and southern Amazon on interdecadal scales (Lee et al., 2011; Marengo, 2004) making it important to examine how these changes propagate into TWS variations over the region. The long-term trends over the entire climatology have been discussed in the third paragraph of Section 3.4 and in Figure 6.

Specific comments:

SC1) Page 2/ line 17: Is it possible to associate the climate and human-induced changes more exactly with the results? What are the driving factors, is it rather the change in climate or the increased human activities that alter the (hydrologic) system in the Amazon?

Response: Hydrological changes in Amazon have been suggested to be a result of a combined impact from climate change and human activities (Cook et al., 2012; Cook and Vizy, 2008; Lee et al., 2011; Malhi et al., 2008; Shukla et al., 1990). Due to the overall scope of the study (focus on droughts) and to keep the manuscript concise, we aimed to keep the Introduction section highly focused on the objective. All the above-mentioned studies, although focus on either climate change or human activities or both, none of them explicitly quantify the causes of the hydrological change, rather it is difficult to do so due to the complex interaction between climate and human activities over a region. Hence, we believe that the overall impact of climate change and human induced changes on the hydrologic system of the Amazon region can be better inferred qualitatively rather than quantitatively. However, we have added the drought related results from these individual studies in the Introduction section.

SC2) Page 2 / lines 23-26: This sentence is in my opinion a repetition of the information presented in the paragraph before (eg. With respect to streamflow reduction). It would be worthwhile considering removing any repetitive statements here and throughout the remaining manuscript.

Response: Thank you for pointing this out. We have removed the sentence and have made some editorial changes on our own for better clarity and flow, ensuring the new paragraphs are logically connected.

SC3) Page 4/ line 6: From the manuscript it does not become clear whether the "opposite trend" between model output and GRACE is only a thing in the Amazon River basin or whether it is issue also on global scale. Please provide this information so readers can get a better idea of the severity of this problem. Also, why is there no explanation available? Maybe provide a brief sentence (based on Scanlon 2018?) since it's bit unsatisfying to read at the moment.

Response: Thank you for this important comment. Scanlon et al. (2018) has shown that most of the global river basins show a different trend behavior in TWS compared to GRACE. In case of the Amazon River basin, models show an opposite trend behavior in TWS. Scanlon et al., (2018) attributes the discrepancy to model shortcomings, such as poor representation of water stores and some hydrological processes, and uncertainty in forcing datasets. Although, their study quantifies the impact of the above-mentioned causes to some extent, the results vary greatly among models and forcing datasets; hence, giving no clear explanation/quantification of the causes of model-GRACE discrepancy. We have added this information about the "opposite trend" between GRACE and model output in the same line. Further, we have also included a reference directing to section 3.3, where we have a detailed discussion regarding the model-GRACE discrepancy and results from Scanlon et al., (2018).

SC4) Page 5/ line 5 "The LHF model": • Even though the paper was already published, adding a flow chart would help the reader to better understand the LHF model and the modelling steps required. • What is the temporal resolution of model output? Please add. • Was the model calibrated? If so, how and using which data and parameters? If not, why not? Please add this information.

Response: Thank you for the suggestion. We have added more information about

the LHF model and its simulation setup in section 2.8. We have prepared a diagram showing the LHF model setup employed in the study; however, we have included this flow diagram in the revised supplementary information because after adding more information about the LHF model and its setup in Section 2.1 and 2.2, we found this to be less important in the main manuscript. LHF model time steps is of 4 minutes but we save the model output at daily intervals. Regarding the third question, we have not calibrated the model using any observed datasets as the land surface, hydrologic and groundwater processes in the model framework are physically based. As such, the entire temporal extent of the model is utilized as validation period rather than dividing it into calibration and validation periods. Although, there are some parameters (e.g., manning's co-efficient) in the physical equations which can be tuned to have a better correlation with observed datasets, we have not performed any tuning as the model represents these processes rather well with the predestined values. We have added this information in Section 2.8 of the revised manuscript.

SC5) Page 5/ line 19: What was the reasoning to use the 2 km version of LHF and not the probably faster 5 km version?

Response: Given the large areas of Amazonian floodplains and its large-scale interaction with the sub-surface water store, it is crucial to simulate these interactions with higher accuracy, to get a relatively complete picture of the hydrology in the region. A finer grid allows the model to retain the spatial details essential for an accurate simulation of floodplain dynamics and the groundwater processes (e.g., convergence along valleys and lateral flow) which are mainly controlled by local topography (Miguez-Macho and Fan, 2012). Moreover, the computational resources which we had at our disposal allowed us to conduct a finer resolution simulation with enough to spare for further analysis. In the past several years, the hydrologic modeling community has put concerted efforts to increase model resolution and move toward hyper-resolution global modeling (grid sizes of 1km or smaller) (Fan et al., 2019; Wood et al., 2011). To contribute to these community efforts LHF has been further refined to 1km grids (Fan

et al., 2017). Thus, we feel that going back to a coarse resolution will be unjust as long as computational facilities permit such high-resolution simulations. We have been using LHF at 5km grids for the continental US where computational cost constraints the grid resolution (Shin et al., 2018). In case of our 5km version, the model domain consisted of Continental United States (CONUS), which is ~3.5 times larger than the domain used in this study, hence forcing us to employ a lower resolution (~5km) over CONUS.

SC6) Page 5/ line 26: It is unclear to me why you perform an extensive spin-up of 150 years but then discard the first year of the modelling period too? Were there issues with model stability or initial conditions? Please explain your choices clearly to avoid any misunderstanding.

Response: Thank you for the thoughtful comment. The spin-up was performed for 150 years with the repeated use of forcing from 1979. All the simulations for 1979 were counted towards spin-up and the main simulation was started from 1980-2015. We did not find any issues with model stability or initial conditions, however, as the manuscript intends to mainly focus on the interdecadal changes, the simulation for year 1979 would anyway have to be discarded, as a single year cannot represent a complete decade. Further, discarding the first year of simulation is also a general methodology adopted in other hydrologic modelling studies. We have added this information in Section 2.8 of the revised manuscript to avoid any misinterpretation.

SC7) Page 6/ line 1 "Atmospheric forcing":   Why did you decide to use the WATCH data as model forcing? Why not the more recent ERA-5 data?   The accuracy of forcing plays an important role, also in the Amazon. The work by Towner et al. (2019, HESSD; https://doi.org/10.5194/hess-2019-44) compares several forcing data sets (ERA-I, ERA-5, and re-forecasts) with respect to their relative impact of model accuracy for the Amazon basin. I think that this could provide a good starting point for a brief discussion about the role of forcing in modelling studies and to explain your decision to use WATCH data.

[Figure]

Response: Thank you for pointing this out. We also agree that the accuracy of forcing plays an important role in the simulating Amazonian hydrology. We decided to use WATCH Forcing Data methodology applied to ERA-Interim reanalysis data (WFDEI) because it has been suggested to well represent the observations and is more suitable for hydrological modelling of Brazilian water resources in the past literature (Monteiro et al., 2016) compared to other datasets considered in the studies. Several bias and atmospheric corrections were also applied in deriving the WFDEI dataset (Weedon et al., 2014), hence making it a widely used forcing dataset for regional and global studies. On the contrary, ERA5 dataset is fairly new and limited studies exist in the literature showing its suitability for hydrological modelling over Amazon. We could have used ERA5 in our analysis and checked its effectiveness over Amazon, however this approach would be vastly different than the overall objective of this study. To further explore the model-GRACE discrepancy, one of the main objectives of this study, we are conducting multiple LHF simulation sets with different forcing datasets; this analysis and the model accuracy with ERA5 dataset will be addressed in our forthcoming paper. Moreover, to better explain our decision of using the WFDEI forcing in our analysis, we have provided additional discussion in Section 2.2 which includes your suggested article of Towner et al., (2019).

SC8) Page 6/ line 10: I understand you use annual input for the land cover? Are you confident that this is acceptable given the model runs at a different temporal resolution I assume? Please add a brief statement why you are convinced your choice made is appropriate.

Response: We use the European Space Agency Climate Change Initiative's Land Cover project (ESA-CCI; http://maps.elie.ucl.ac.be/CCI/) land cover maps to represent the land use land cover (LULC) dynamics in our model framework. Land cover dataset from ESA-CCI is the only dataset currently available which satisfies the LHF model requirements in terms of spatial extent, temporal extent and spatial resolution. Although, one can generate land cover maps at higher temporal resolution using Land-

sat imagery, the resulting process will greatly differ from the overall scope of this study. Moreover, we believe that the impacts of LULC change on the regional hydrology dominate on a longer time scale and its seasonal dynamics are well captured through LAI changes. Both of these dynamics are incorporated in the framework with an annual input of land cover and monthly LAI input. Further, usage of annual LULC input is also the general practice in hydrologic impact studies. To facilitate a better understanding of the model setup, we have added these additional details regarding the use of annual land cover input in Section 2.3 of the revised manuscript.

SC9) Page 6/ line 19: how did you decide to use LAI value 5 as threshold for transitioning into forest? Is this based on expert knowledge, scientific literature or just an arbitrary decision? I can image that in vegetation-rich areas (ie. With generally high LAI values) such as the Amazon this threshold can have a marked influence on the total area eventually specified as forest.

Response: Thank you for pointing out this issue. The threshold of LAI=5 for the forest transition is based on scientific literature. The threshold value was mainly based on the study conducted by Asner et al., (2003) which presents a synthesis of global LAI values for different land cover types. Asner et al., (2003) showed that the evergreen broadleaf and needleleaf forests, which are the major forest types in Amazon, have average LAI values greater than 5 (5.8 and 6.7, respectively). Other studies also classify the evergreen forests in the same LAI range (Myneni et al., 2007; Xu et al., 2018); for example, Myneni et al., (2007) studied the seasonal swings in LAI values and showed that the mean annual LAI is ~5 over the entire Amazonian rainforest (Figure 1A of the citation). Hence, we used the threshold of LAI=5 to get a first-hand approximation of the past forest cover in Amazon. To elaborate a bit more on the method we used to back extrapolate land cover, we have added more information in Section 2.3.

SC10) Page 7/ line 11: did you use monthly averages? Please add this information.

Response: We used the observed monthly averaged streamflow data from Agência Nacional de Águas (ANA) in Brazil for model validation. This additional information about the observed streamflow data is added in Section 2.4.1 of the revised manuscript.

SC11) Page 7/ line 14: Do all stations cover the same period or not? And what about missing values in the time series – are there any and if so how did you treat them?

Response: The streamflow stations were selected based on their data length and data gaps. Data periods varied among streamflow stations, with some spanning over the entire modelling period (i.e. 36 years) and others for more than 30 years. As the observed streamflow values are solely used for model validation, the months with missing data were skipped from the statistical analysis. More information about the observed streamflow data and their usage in the analysis is added in Section 2.4.1 of the revised manuscript.

SC12) Page 9/ line 29: Why is it that GRACE shows little agreement in small basins? Please add a brief explanation. Also, I am wondering whether using GRACE is then the right approach for those basins – would it not be an alternative to skip the GRACE comparison for small basins where it's known a priori that agreement will be small?

Response: GRACE error mentioned in this manuscript refers to bias and leakage correction errors (Landerer and Swenson, 2012; Longuevergne et al., 2010). This type of GRACE error is mainly dependent on the basin size, which increases with a decrease in basin size (Longuevergne et al., 2010). We use the GRACE data to ensure that the simulated TWS variations are within the plausible limits for the Amazon basin and its sub-basins. Also, one of the key discussions made in the manuscript is the discrepancy between model and GRACE, hence skipping GRACE comparison for smaller basins would result into an incomplete analysis. Moreover, to tackle the leakage errors associated with GRACE spherical harmonics products, we also employ the GRACE mascon products in our analysis, which are known to better capture the TWS signal by reducing the error from leakage (Save et al., 2016; Scanlon et al., 2016). Even though,

some of the Amazonian sub-basins are fairly small compared to their neighbors, the smallest river basin (i.e. Japura, ∼256,000 km2) under consideration still has a basin area higher than the GRACE footprint (∼200,000 km2) (Longuevergne et al., 2010). Hence, skipping the model-GRACE comparison for these basins will not be wise, as it will not only create a void in the validation but also introduce inconsistency (or even doubt to the reader on why they are excluded) with other analyses done in this study. We have thus decided to include those smaller basins in the analyses.

SC13) Page 11/ line 7: I have my doubts whether the manuscript presents a "state-of-the-art" framework. After all, the model used is already available for quite a while and the main novelty of the presented work is the extensive analysis with GRACE and streamflow data for the Amazon and its sub-basins for a long period of time (which is an important contribution to current process understanding!). Another example would be the WATCH forcing which could be updated with more recent data sets. If you're convince the framework is nevertheless state-of-the-art, I would like to see an elaboration in the model description section why that is the case.

Response: Thank you for pointing this out. We have removed the term "state-of-the-art" from the specified line. We agree that the LHF model has been available for quite a while now, but we cannot stress less on the role of the simulation setup for making the overall framework "state-of-the-art". The version of LHF model we use in this study was last updated in 2012. Novelty of the LHF model lies in that incorporates sophisticated land surface process such as a prognostic groundwater store, dynamic water table and its interaction with surface water stores, lateral groundwater exchange, sea-level influence on coastal drainage and river-floodplain routing by resolving the full momentum equation of open channel flow. In this study, we further incorporated hydrological interactions with human activities, such as LULC change, through dynamic land cover input at annual scale (see response for SC8) and leaf area index input at a monthly scale, hence creating a comprehensive framework for assessing long-term hydrological changes. Although, many other comparable hydrological models also represent some of the above-mentioned hydrological processes in their modelling framework, most of them lack the explicit groundwater scheme (e.g., NOAH, VIC) (Ek et al., 2003; Liang et al., 1994)and the flood dynamics in their framework are fairly simplified, essentially making the surface-subsurface interaction mostly linear reservoir based (e.g., WGHM, PCR-GLOBWB) (Alcamo et al., 2003; van Beek and Bierkens, 2009). Similar level of detail as LHF, is also found in the CaMa-Flood (Yamazaki et al., 2011) model which incorporates a river-floodplain routing system; however, the scheme has not integrated the Saint-Venant equation with other land-surface hydrological processes till date, hence demoting it in the hierarchy. Therefore, the evident contribution from the long-term simulation setup incorporating the dynamic human role in impacting the hydrological cycle in addition to the original novelty of the LHF model are the key factors behind highlighting our framework as "state-of-the-art". To demonstrate this to our readers, we have added more information in respective sections of the revised manuscript. For details regarding the usage of WATCH forcing data in this study, please refer the response for SC7.

SC14) Page 14/ line 3: What was your motivation to employ HydroSHEDS basins for this specific analysis? Do these "sub-catchments" match the geographical extent of the sub-basins you are referring to in the remainder of the manuscript? If not, this choice somehow complicates the analysis by introducing another geographical unit; in this case, I would advise to stick to the sub-basin definition used for the other analyses.

Response: Yes, the "sub-catchments" exactly match the geographical extent of the sub-basins referred in the rest of the manuscript. The LHF model also utilize the topography information from the HydroSHEDS dataset. We decided to use the "sub-catchments", which are essentially the sub-basins of the Amazonian sub-basins, to study the hydrological drought propagation at higher spatial resolution. This allows us to take advantage of the wide number of the streamflow estimates obtained from the model, to conduct an in-depth analysis of the complex interaction between LULC changes and streamflow. Analyzing the hydrological drought trends at a sub-basin level

will constrain the analysis at merely 8 points, further making it difficult to understand while severely underestimating the role of LULC in governing streamflow generated from the region. Therefore, even though we agree with the reviewer that we could use sub-basins for the sake of consistency but going down to smaller catchments would provide finer details of the role of LULC in regional hydrology.

SC15) Page 15/ line 3: You are mentioning "important insights" but I don't see any further elaboration what those insights could be. Please append this information!

Response: Thank you for the suggestion. We have moved the statement at a location which more appropriately justifies the term "important insights". The statement now is situated before the start of the discussion in Section 3.6 trailed by the detailed description of individual insight it infers.

SC16) Page 15/ line 4 "Intensification of the Amazonian Dry Season": This chapter could profit from discussing findings from other literature to put your results into perspective. Please add where applicable!

Response: Thank you for the suggestion. We have added a more exhaustive discussion regarding the "Intensification of Amazonian Dry Season" by combining the results from previous literature and this study in Section 3.7.

References Alcamo, J., Döll, P., Henrichs, T., Kaspar, F., Lehner, B., Rösch, T. and Siebert, S.: Development and testing of the WaterGAP 2 global model of water use and availability, Hydrol. Sci. J., 48(3), 317–337, 2003.

Asner, G. P., Scurlock, J. M. O. and A. Hicke, J.: Global synthesis of leaf area index observations: implications for ecological and remote sensing studies, Glob. Ecol. Biogeogr., 12(3), 191–205, doi:10.1046/j.1466-822X.2003.00026.x, 2003. van Beek, L. P. H. and Bierkens, M. F. P.: The global hydrological model PCR‐GLOBWB: Conceptualization, parameterization and verification report, Dep. Phys. Geogr., Utr. Univ., Utrecht, Netherlands, 2009.

Cook, B., Zeng, N. and Yoon, J. H.: Will Amazonia dry out? Magnitude and causes of change from IPCC climate model projections, Earth Interact., 16(3), doi:10.1175/2011EI398.1, 2012.

Cook, K. H. and Vizy, E. K.: Effects of twenty-first-century climate change on the Amazon rain forest, J. Clim., 21(3), 542–560, doi:10.1175/2007JCLI1838.1, 2008.

Ek, M. B., Mitchell, K. E., Lin, Y., Rogers, E., Grunmann, P., Koren, V., Gayno, G. and Tarpley, J. D.: Implementation of Noah land surface model advances in the National Centers for Environmental Prediction operational mesoscale Eta model, J. Geophys. Res. Atmos., 108(D22), doi:10.1029/2002JD003296, 2003.

Fan, Y., Miguez-Macho, G., Jobbágy, E. G., Jackson, R. B. and Otero-Casal, C.: Hydrologic regulation of plant rooting depth, Proc. Natl. Acad. Sci., 114(40), 10572–10577, doi:10.1073/pnas.1712381114, 2017.

[revised manuscript text omitted]

---

## Author Comment (AC2) · 4 Jun 2019

This study applied a physics-based hydrological model and GRACE product to investigate the hydrological changes in the Amazon basin, especially the water storage and how it related to droughts, for 36 years period. The results of this study are comprehensive and the findings are significant, which improve the understanding of hydrology in Amazon. But there are still some concerns in the manuscript need to be addressed.

Response: Thank you for your positive evaluation of the manuscript. We found significant improvement in the quality of the manuscript following your and the other reviewer's comments. Below we provide detailed responses to your comments along with the references in necessary locations. Please note that following the first reviewer's comment, supplementary figure numbers have changed in the revised manuscript.

GC1) The first two questions regard the modeling approaches. Firstly, it was mentioned that the atmospheric forcing data are spatially interpolated using a bilinear interpolation method to the model grid. The issue is, for example, rainfall events are usually local and spatially discontinuous, whether the bilinear interpolation is appropriate for some of the climatology data.

Response: The Leaf-Hydro-Flood (LHF) model version we used in this study interpolates WFDEI forcing data from 0.5 degrees to the 1 arc minute model grid (~2km) using bilinear interpolation. We agree that more sophisticated interpolation techniques, such as kriging, yield more accurate results compared to the bilinear approach, especially with the rainfall data. However, these sophisticated methods also come at a significant computational cost. Moreover, several previous studies (Fan et al., 2017; Miguez-Macho and Fan, 2012; Pokhrel et al., 2013, 2014) along with this study have shown that even with the bilinear interpolation LHF model yield accurate results. Hence, we firmly believe that implementing a more sophisticated interpolation technique would be unnecessary given the current accuracy achieved from the model. Further it would add significant computational burden as the interpolation is done within the model over a very large domain (~4 million grids).

GC2) Secondly, regarding the LULC change applied to the model, LAI higher than 5 are considered as forest canopy. Then the question is, how does this approach deal with the seasonal variation of LAI as for LULC change?

Response: Thank you for pointing out this issue. The threshold of LAI=5 for the forest transition is the mean annual LAI calculated by aggregating the 8-day composites from GLASS data for every year. We use this mean annual LAI estimates only for deriving

the 1980-1991 (years not included in the ESA-CCI data) annual land cover maps. The seasonal variations in LAI are separately incorporated in the model framework at a monthly scale. To elaborate a bit more on the method we used to back extrapolate land cover, we have added more information in Section 2.3 of the revised manuscript. Further, the threshold value was mainly based on the study conducted by Asner et al., (2003) which presents a synthesis of global LAI values for different land cover types. Asner et al., (2003) showed that the evergreen broadleaf and needleleaf forests, which are the major forest types in Amazon, have average LAI values greater than 5 (5.8 and 6.7, respectively). Other studies also classify the evergreen forests in the same LAI range (Myneni et al., 2007; Xu et al., 2018); for example, Myneni et al., (2007) studied the seasonal swings in LAI values and showed that the mean annual LAI is ∼5 over the entire Amazonian rainforest (Figure 1A of the citation). Hence, we used the threshold of LAI=5 to get a first-hand approximation of the past forest cover in Amazon.

GC3) The manuscript consists of 5 parts, but the model descriptions in Section 2 should belong to Section 3, methods. Thus, it would be better to re-organize the contents and the structure of the manuscript.

Response: Thank you for the suggestion. We agree that moving the model descriptions to Section 3, methods, will result into a better structure. We thought of improving it further by combining the current sections 2 and 3 together. In the revised manuscript, we have revised the structure as follows, 2. Model, Data and Methods 2.1 The Leaf-Hydro-Flood (LHF) model 2.2 Atmospheric Forcing 2.3 Land Use Land Cover and Leaf Area Index 2.4 Validation Data 2.4.1 Observed Streamflow 2.4.2 GRACE Data 2.5 TWS Drought Severity Index 2.6 Occurrence and Duration of Drought 2.7 Dry Season Total Water Deficit 2.8 Simulation Setup

GC4) In addition, Figures S3, S6, and S8 are not referred nor discussed in the manuscript. Moreover, there are also some specific comments as below.

Response: Thank you for pointing this out. We have added supplementary figure

references for the in required locations in the revised manuscript.

SC1. P3L13-15, some of these 'more recent' literature are still more than 10 years old. The author should cite some real more recent papers.

Response: Thank you for the suggestion. We have removed the old citations and have added more recent literature in the revised manuscript. New literature added to the revised manuscript consists of studies conducted in 2010s such as Fan et al., (2019), Shin et al., (2018) and Wang et al., (2019).

SC2. P8L27-29, the description of the symbols in the figure should also be presented in the figure caption.

Response: Thank you for the suggestion. We have added the symbol descriptions in the Figure 1 caption. Also, please note that, according to the other reviewer's comment we have removed the figure description from the main text.

SC3. P9L7, this conclusion is not easy to clarify from the figures. Please describe more clearly and specifically.

Response: Thank you for the suggestion. River basins, such as Japura and Negro, are characterized by high topographic gradients, resulting into an uneven seasonal stream-flow pattern. These gradients are not adequately represented in the model framework due to the limitation in model resolution, hence causing higher discrepancies with the observed values. We have added this information in a concise manner in the revised manuscript to have a better understanding of the conclusion inferred from Figure S2.

SC4. P9L14, the discrepancies in some basins cannot be seen from Figure S2, for example, by which metrics?

Response: Thank you for pointing this out. We specifically wanted to point out the discrepancies in the simulated and observed magnitude of peaks in seasonal streamflow cycle. Xingu, Tapajos and Tocantins sub-basins show significant differences between the simulated and observed seasonal peak of streamflow (smaller right panels of each

basin, Figure S3 of the revised manuscript) due to the higher hydropower activity compared to the other river basins. We have edited the statement by including additional information to avoid confusion.

SC5. P12L13, the method of t-test should be described in the methodology section unless it is an ordinary t-test.

Response: The t-test methodology we used is the ordinary t-test. We decided to skip its description from the methodology section as the test is very commonly used.

SC6. P14L14, it should be 'Figure 10'.

Response: Thank you for pointing this mistake. We have corrected it to "Figure 10".

SC7. Figure 5, the color change of the rivers is not clear. The line widths of the rivers should be increased.

Response: Figure 5 shows the interdecadal difference in TWS components at the original model resolution (∼2km). The data presented in the figure is gridded data, hence we cannot represent the rivers in a polyline format. For better visualization we have removed the inland water and country borders.

SC8. Figure S1 lacks the north arrow and the scale. Moreover, the author should mark all sub-basins and major rivers in this figure.

Response: Thank you for the suggestion. We have marked the sub-basin borders in Figure S2 of the revised manuscript; however, we believe that adding the scale and north arrow to the figure would become redundant due to the presence of the geographical co-ordinates.

SC9. Figure 7, y-axis label is missing.

Response: Thank you for the suggestion. We have added a y-axis label in Figure 5.

SC10. It would be better to include geo-coordinates for all spatial plots, e.g., Figure 3,

4, 5, 6, 8, 9, S4, S5, S7, and S9.

Response: Thank you for the suggestion. We have made the suggested changes in the figures.

References Asner, G. P., Scurlock, J. M. O. and A. Hicke, J.: Global synthesis of leaf area index observations: implications for ecological and remote sensing studies, Glob. Ecol. Biogeogr., 12(3), 191–205, doi:10.1046/j.1466-822X.2003.00026.x, 2003.

Fan, Y., Miguez-Macho, G., Jobbágy, E. G., Jackson, R. B. and Otero-Casal, C.: Hydrologic regulation of plant rooting depth, Proc. Natl. Acad. Sci., 114(40), 10572–10577, doi:10.1073/pnas.1712381114, 2017.

Fan, Y., Clark, M., Lawrence, D. M., Swenson, S., Band, L. E., Brantley, S. L., Brooks, P. D., Dietrich, W. E., Flores, A., Grant, G., Kirchner, J. W., Mackay, D. S., McDonnell, J. J., Milly, P. C. D., Sullivan, P. L., Tague, C., Ajami, H., Chaney, N., Hartmann, A., Hazenberg, P., McNamara, J., Pelletier, J., Perket, J., Rouholahnejad-Freund, E., Wagener, T., Zeng, X., Beighley, E., Buzan, J., Huang, M., Livneh, B., Mohanty, B. P., Nijssen, B., Safeeq, M., Shen, C., van Verseveld, W., Volk, J. and Yamazaki, D.: Hillslope Hydrology in Global Change Research and Earth System Modeling, Water Resour. Res., 55(2), 1737–1772, doi:10.1029/2018WR023903, 2019.

Miguez-Macho, G. and Fan, Y.: The role of groundwater in the Amazon water cycle: 1. Influence on seasonal streamflow, flooding and wetlands, J. Geophys. Res. Atmos., 117(15), 1–30, doi:10.1029/2012JD017539, 2012.

Myneni, R. B., Yang, W., Nemani, R. R., Huete, A. R., Dickinson, R. E., Knyazikhin, Y., Didan, K., Fu, R., Negron Juarez, R. I., Saatchi, S. S., Hashimoto, H., Ichii, K., Shabanov, N. V., Tan, B., Ratana, P., Privette, J. L., Morisette, J. T., Vermote, E. F., Roy, D. P., Wolfe, R. E., Friedl, M. A., Running, S. W., Votava, P., El-Saleous, N., Devadiga, S., Su, Y. and Salomonson, V. V.: Large seasonal swings in leaf area of Amazon rainforests, Proc. Natl. Acad. Sci., 104(12), 4820–4823, doi:10.1073/pnas.0611338104,

**HESSD**
[Figure]

2007.

Pokhrel, Y. N., Fan, Y., Miguez-Macho, G., Yeh, P. J. F. and Han, S. C.: The role of groundwater in the Amazon water cycle: 3. Influence on terrestrial water storage computations and comparison with GRACE, J. Geophys. Res. Atmos., 118(8), 3233–3244, doi:10.1002/jgrd.50335, 2013.

Pokhrel, Y. N., Fan, Y. and Miguez-Macho, G.: Potential hydrologic changes in the Amazon by the end of the 21st century and the groundwater buffer, Environ. Res. Lett., 9(8), 084004, doi:10.1088/1748-9326/9/8/084004, 2014.

Shin, S., Pokhrel, Y. and Miguez-Macho, G.: High Resolution Modeling of Reservoir Release and Storage Dynamics at the Continental Scale, Water Resour. Res., 55, doi:10.1029/2018WR023025, 2018.

Wang, K., Shi, H., Chen, J. and Li, T.: An improved operation-based reservoir scheme integrated with Variable Infiltration Capacity model for multiyear and multipurpose reservoirs, J. Hydrol., 571, 365–375, doi:10.1016/j.jhydrol.2019.02.006, 2019.

Xu, B., Park, T., Yan, K., Chen, C., Zeng, Y., Song, W., Yin, G., Li, J., Liu, Q., Knyazikhin, Y. and Myneni, R. B.: Analysis of Global LAI/FPAR Products from VIIRS and MODIS Sensors for Spatio-Temporal Consistency and Uncertainty from 2012–2016, Forests, 9(2), doi:10.3390/f9020073, 2018.